# KERNEL AND RICH REGIMES IN OVERPARAMETRIZED MODELS

## ABSTRACT

A recent line of work studies overparametrized neural networks in the "kernel regime," i.e. when the network behaves during training as a kernelized linear predictor, and thus training with gradient descent has the effect of finding the minimum RKHS norm solution. This stands in contrast to other studies which demonstrate how gradient descent on overparametrized multilayer networks can induce rich implicit biases that are not RKHS norms. Building on an observation by Chizat and Bach (2018), we show how the ***scale of the initialization*** controls the transition between the "kernel" (aka lazy) and "rich" (aka active) regimes and affects generalization properties in multilayer homogeneous models. We provide a complete and detailed analysis for a simple two-layer model that already exhibits an interesting and meaningful transition between the kernel and rich regimes, and we demonstrate the transition for more complex matrix factorization models and multilayer non-linear networks.

## 1 INTRODUCTION

A string of recent papers study neural networks trained with gradient descent in the "kernel regime." They observe that, in a certain regime, networks trained with gradient descent behave as kernel methods (Jacot et al., 2018; Daniely et al., 2016; Daniely, 2017). This allows one to prove convergence to zero error solutions in overparametrized settings (Du et al., 2018; 2019; Allen-Zhu et al., 2018), and also implies gradient descent will converge to the minimum norm solution (in the corresponding RKHS) (Chizat and Bach, 2018; Arora et al., 2019b; Mei et al., 2019) and more generally that models inherit the inductive bias and generalization behavior of the RKHS. This suggests that, in a certain regime, deep models can be equivalently replaced by kernel methods with the "right" kernel, and deep learning boils down to a kernel method with a fixed kernel determined by the architecture and initialization, and thus it can only learn problems learnable by some kernel.

This contrasts with other recent results that show how in deep models, including infinitely overparametrized networks, training with gradient descent induces an inductive bias that cannot be represented as an RKHS norm. For example, analytic and/or empirical results suggest that gradient descent on deep linear convolutional networks implicitly biases toward minimizing the $L_p$ bridge penalty, for $p = 2/\text{depth} \leq 1$, in the frequency domain (Gunasekar et al., 2018b); weight decay on an infinite width single input ReLU implicitly biases towards minimizing the second order total variations $\int |f''(x)|\, dx$ of the learned function (Savarese et al., 2019); and gradient descent on a overparametrized matrix factorization, which can be thought of as a two layer linear network, induces nuclear norm minimization of the learned matrix (Gunasekar et al., 2017) and can ensure low rank matrix recovery (Li et al., 2018). All these natural inductive biases ($L_p$ bridge penalty for $p < 1$, total variation norm, nuclear norm) are not Hilbert norms, and therefore *cannot* be captured by a kernel. This suggests that training deep models with gradient descent can behave very differently from kernel methods, and have much richer inductive biases.

One might then ask whether the kernel approximation indeed captures the behavior of deep learning in a relevant and interesting regime, or does the success of deep learning come when learning escapes this regime? In order to understand this, we must first carefully understand when each of these regimes hold, and how the transition between the "kernel" regime and the "rich" regime happens.

Some investigations of the kernel regime emphasized the number of parameters ("width") going to infinity as leading to this regime. However Chizat and Bach (2018) identified the scale of the model

as a quantity controlling entry into the kernel regime. Their results suggest that for any number of parameters (any width), a model can be approximated by a kernel when its scale at initialization goes to infinity (see details in Section 3). Considering models with increasing (or infinite) width, the relevant regime (kernel or rich) is determined by how the scaling at initialization behaves as the width goes to infinity. In this paper we elaborate and expand of this view, carefully studying how the scale of initialization effects the model behaviour for $D$-homogeneous models.

In Section 4 we provide a complete and detailed study for a simple 2-homogeneous model that can be viewed as linear regression with squared parametrization, or as a "diagonal" linear neural network. For this model we can exactly characterize the implicit bias of training with gradient descent, as a function of the scale $\alpha$ of initialization, and see how this implicit bias becomes the $\ell_2$ norm in the $\alpha \to \infty$ kernel regime, but the $\ell_1$ norm in the $\alpha \to 0$ rich regime. We can therefore understand how, e.g. for a high dimensional sparse regression problem, where it is necessary to discover the relevant features, we can get good generalization when the initialization scale $\alpha$ is small, but not when $\alpha$ is large. Deeper networks corresponds to higher orders of homogeneity, and so in Section 5 we extend our study to a $D$-homogeneous model, studying the effects of $D$. In Sections 6 and 7, we demonstrate similar transitions experimentally in matrix factorization and non-linear networks.

## 2 Setup and preliminaries

We consider models $f : \mathbb{R}^p \times \mathcal{X} \to \mathbb{R}$ which map parameters $\mathbf{w} \in \mathbb{R}^p$ and examples $\mathbf{x} \in \mathcal{X}$ to predictions $f(\mathbf{w}, \mathbf{x}) \in \mathbb{R}$. We denote the predictor implemented by the parameters $\mathbf{w}$ as $h_\mathbf{w} = F(\mathbf{w})$ such that $h_\mathbf{w}(\mathbf{x}) = f(\mathbf{w}, \mathbf{x})$. Much of our focus will on models, such a linear networks, which are linear in $\mathbf{x}$ (but not on the parameters $\mathbf{w}$!), in which case $F(\mathbf{w}) \in \mathcal{X}^*$ is a linear predictor and can be represented as a vector $\boldsymbol{\beta}_\mathbf{w}$ with $f(\mathbf{w}, \mathbf{x}) = \langle \boldsymbol{\beta}_\mathbf{w}, \mathbf{x} \rangle$. Such models are essentially alternate parametrizations of linear models, but as we shall see that change of parametrization is crucial.

In this paper, we consider models that are $D$-positive homogeneous in the parameters $\mathbf{w}$, for some integer $D \geq 1$, meaning that for any $c \in \mathbb{R}_+$, $F(c \cdot \mathbf{w}) = c^D F(\mathbf{w})$ and $f(c \cdot \mathbf{w}, \mathbf{x}) = c^D f(\mathbf{w}, \mathbf{x})$. We refer to such models simply as $D$-homogeneous. Many interesting model classes have this property, including multi-layer ReLU networks with fully connected and convolutional layers, layered linear neural networks, and matrix factorization where $D$ corresponds to the depth of the network.

Consider a training set $\{(\mathbf{x}^{(n)}, y^{(n)})\}_{n=1}^N$ consisting of $N$ examples of input label pairs. For a given loss function $\ell : \mathbb{R} \times \mathbb{R} \to \mathbb{R}$, the loss of the model parametrized by $\mathbf{w}$ is $L(\mathbf{w}) = L(F(\mathbf{w})) = \sum_{n=1}^N \ell(f(\mathbf{w}, \mathbf{x}^{(n)}), y^{(n)})$. We will focus mostly on the squared loss $\ell_{\text{sq}}(\hat{y}, y) = (\hat{y} - y)^2$. We slightly abuse notation and use $f(\mathbf{w}, X) \in \mathbb{R}^N$ to denote the vector of predictions $[f(\mathbf{w}, \mathbf{x}^{(1)}), \dots, f(\mathbf{w}, \mathbf{x}^{(N)})]$ and so for the squared loss we can write $L(\mathbf{w}) = \|f(\mathbf{w}, X) - \mathbf{y}\|_2^2$, where $\mathbf{y} \in \mathbb{R}^N$ is the vector of target labels.

Minimizing the loss $L(\mathbf{w})$ using gradient descent amounts to iteratively updating the parameters

$$\mathbf{w}(k+1) = \mathbf{w}(k) - \eta \nabla L(\mathbf{w}(k)). \tag{1}$$

We consider gradient descent with infinitesimally small stepsize $\eta$, i.e. gradient flow dynamics

$$\dot{\mathbf{w}}(t) = -\nabla L(\mathbf{w}(t)). \tag{2}$$

We are particularly interested in the scale of initialization and we capture it through a scalar parameter $\alpha \in \mathbb{R}_+$. For scale $\alpha$, we will denote by $\mathbf{w}_\alpha(t)$ the gradient flow path (2) with the initial condition $\mathbf{w}_\alpha(0) = \alpha \mathbf{w}_0$ for some fixed $\mathbf{w}_0$. We use $h_\alpha(t) = F(\mathbf{w}_\alpha(t))$, and for linear predictors $\boldsymbol{\beta}_\alpha(t) = \boldsymbol{\beta}_{\mathbf{w}_\alpha(t)}$, to denote the dynamics on the predictor $F(\mathbf{w})$ induced by the gradient flow on $\mathbf{w}$.

In many cases, we expect the dynamics to converge to a minimizer of $L(\mathbf{w})$, though proving this happens will not be our main focus. Rather, we are interested in the underdetermined case, $N \ll p$, where there are generally many minimizers of $L(\mathbf{w})$, all with $f(\mathbf{w}, X) = \mathbf{y}$ and $L(\mathbf{w}) = 0$. Our main focus is which of the many minimizers does gradient flow converge to. That is, we want to characterize $\mathbf{w}_\alpha(\infty) = \lim_{t \to \infty} \mathbf{w}_\alpha(t)$ or, more importantly, the predictor $h_\alpha(\infty) = F(\mathbf{w}_\alpha(\infty))$ or $\boldsymbol{\beta}_\alpha(\infty) = \boldsymbol{\beta}_{\mathbf{w}_\alpha(\infty)}$ we converge to, and how these depend on the scale $\alpha$. In underdetermined problems, where there are many zero error solutions, simply fitting the data using the model does not provide enough inductive bias to ensure generalization. But in many cases, the specific solution

reached by gradient flow (or some other optimization procedure) has special structure, or minimizes some implicit regularizer, and this structure or regularizer provides the needed inductive bias (Gunasekar et al., 2018b;a; Soudry et al., 2018; Ji and Telgarsky, 2018).

## 3 THE KERNEL REGIME

Gradient descent/flow considers only the first-order approximation of the model w.r.t. $\mathbf{w}$:

$$f(\mathbf{w}, \mathbf{x}) = f(\mathbf{w}(t), x) + \langle \mathbf{w} - \mathbf{w}(t), \nabla_{\mathbf{w}} f(\mathbf{w}(t), \mathbf{x}) \rangle + O(\|\mathbf{w} - \mathbf{w}(t)\|^2). \tag{3}$$

That is, locally around any $\mathbf{w}(t)$, gradient flow operates on the model as if it were an affine model $f(\mathbf{w}, \mathbf{x}) \approx f_0(\mathbf{x}) + \langle \mathbf{w}, \phi_{\mathbf{w}(t)}(\mathbf{x}) \rangle$ with feature map $\phi_{\mathbf{w}(t)}(\mathbf{x}) = \nabla_{\mathbf{w}} f(\mathbf{w}(t), \mathbf{x})$, corresponding to the *tangent kernel* $K_{\mathbf{w}(t)}(x, x') = \langle \nabla_{\mathbf{w}} f(\mathbf{w}(t), \mathbf{x}), \nabla_{\mathbf{w}} f(\mathbf{w}(t), \mathbf{x}) \rangle$ (Jacot et al., 2018; Zou et al., 2018; Yang, 2019; Lee et al., 2019). Of particular interest is the tangent kernel at initialization, $K_{\mathbf{w}_\alpha(0)} = \alpha^{2(D-1)} K_0$ where we denote $K_0 = K_{\mathbf{w}_0}$.

The "kernel regime" refers to a situation in which the tangent kernel $K_{\mathbf{w}(t)}$ does not change over the course of optimization, and less formally to the regime where it does not change significantly, i.e. where $\forall_t K_{\mathbf{w}(t)} \approx K_{\mathbf{w}(0)}$. In this regime, training the model is exactly equivalent to training an affine model $\tilde{f}(\mathbf{w}, \mathbf{x}) = \alpha^D f(\mathbf{w}_0, \mathbf{x}) + \langle \mathbf{w}, \alpha^{D-1} \phi_0(\mathbf{x}) \rangle$ with kernelized gradient descent/flow with the kernel $\alpha^{2(D-1)} K_0$ and a "bias term" of $\alpha^D f(\mathbf{w}_0, \mathbf{x})$. To avoid handling this bias term, and in particular its scaling, Chizat and Bach (2018) suggest using "unbiased" initializations such that $F(\mathbf{w}_0) = 0$, so that the bias term vanishes. This can often be achieved by replicating units or components with opposite signs at initialization, which is the approach we use here (see Sections 4–6 for examples and details).

For underdetermined problem with multiple solutions $f(\mathbf{w}, X) = \mathbf{y}$, unbiased kernel gradient flow (or gradient descent) converges to the minimum norm solution $\hat{h}_K = \arg\min_{h(X)=\mathbf{y}} \|h\|_K$, where $\|h\|_K$ is the RKHS norm corresponding to the kernel. And so, in the kernel regime, we will have that $h(\infty) = \hat{h}_{K_0}$, and the implicit bias of training is precisely given by the kernel.

When does the "kernel regime" happen? Chizat and Bach (2018) showed that for any homogeneous[1] model satisfying some technical conditions, the kernel regime is reached as $\alpha \to \infty$. That is, as we increase the scale of initialization, the dynamics converge to the kernel gradient flow dynamics with the kernel $K_0$, and we have $\lim_{\alpha \to \infty} h_\alpha(\infty) = \hat{h}_K$. In Sections 4 and 5 we prove this limit directly for our specific models, and we also demonstrate it empirically for matrix factorization and deep networks in Sections 6 and 7.

In contrast, and as we shall see in later sections, the $\alpha \to 0$ small initialization limit often leads to a very different and rich inductive bias, e.g. inducing sparsity or low-rank structure (Gunasekar et al., 2017; Li et al., 2018; Gunasekar et al., 2018b), that allows for generalization in many settings where kernel methods would not. We refer to this limit reached as $\alpha \to 0$ as the "rich regime." This regime is also referred to as the "active" or "adaptive" regime (Chizat and Bach, 2018) since the tangent kernel $K_{\mathbf{w}(t)}$ changes over the course of training, in a sense adapting to the data. We argue that this regime is the one that truly allows us to exploit the power of depth, and thus is the more relevant regime for understanding the success of deep learning.

## 4 DETAILED STUDY OF A SIMPLE DEPTH-2 MODEL

We study in detail a simple 2-homogeneous model. Consider the class of linear functions over $\mathcal{X} = \mathbb{R}^d$, with squared parameterization as follows:

$$f(\mathbf{w}, \mathbf{x}) = \sum_{i=1}^d (\mathbf{w}_{+,i}^2 - \mathbf{w}_{-,i}^2) \mathbf{x}_i = \langle \boldsymbol{\beta}_{\mathbf{w}}, \mathbf{x} \rangle, \text{ where } \mathbf{w} = \begin{bmatrix} \mathbf{w}_+ \\ \mathbf{w}_- \end{bmatrix} \in \mathbb{R}^{2d} \text{ and } \boldsymbol{\beta}_{\mathbf{w}} = \mathbf{w}_+^2 - \mathbf{w}_-^2 \tag{4}$$

---

[1]Chizat and Bach did not consider only homogenous models, and instead of studying the scale of initialization they studied scaling the output of the model. For homogeneous models, the dynamics obtained by scaling the initialization are equivalent to those obtained by scaling the output, and so here we focus on homogenous models and on scaling the initialization.

where we use the notation $\mathbf{z}^2$ for $\mathbf{z} \in \mathbb{R}^d$ to denote elementwise squaring. We consider initializing all weights equally with $\mathbf{w}_0 = \mathbf{1}$.

This is nothing but a linear regression model, except with an unconventional parametrization. The model can also be thought of as a "diagonal" linear neural network (i.e. where the weight matrices have diagonal structure) with $2d$ units. A standard diagonal linear network would have $d$ units, with each unit connected to just a single input unit with weights $u_i$ and the output with weight $v_i$, thus implementing the model $f((\boldsymbol{u}, \boldsymbol{v}), \mathbf{x}) = \sum_i \boldsymbol{u}_i \boldsymbol{v}_i \mathbf{x}_i$. But if at initialization $|\boldsymbol{u}_i| = |\boldsymbol{v}_i|$, their magnitude will remain equal and their signs will not flip throughout training, and so we can equivalently replace both with a single weight $\mathbf{w}_i$, yielding the model $f(\mathbf{w}, \mathbf{x}) = \left\langle \mathbf{w}^2, \mathbf{x} \right\rangle$.

The reason for using both $\mathbf{w}_+$ and $\mathbf{w}_-$ (or $2d$ units) is two-fold. First, it ensures that the image of $F(\mathbf{w})$ is all (signed) linear functions, and thus the model is truly equivalent to standard linear regression. Second, it allows for initialization at $F(\alpha \mathbf{w}_0) = 0$ without this being a saddle point from which gradient flow will never escape.[2]

The model (4) is perhaps the simplest non-trivial $D$-homogeneous model for $D > 1$, and we chose it for this reason, as it already exhibits distinct and interesting kernel and rich regimes. Furthermore, we can completely understand both the implicit regularization driving this model and the transition between the regimes analytically.

Consider the behavior of the limit of gradient flow (2) as a function of the initialization, in the underdetermined $N \ll d$ case where there are many possible solutions $X\boldsymbol{\beta} = \mathbf{y}$. The tangent kernel at initialization is $K_0(\mathbf{x}, \mathbf{x}') = 8\alpha^2 \left\langle \mathbf{x}, \mathbf{x}' \right\rangle$, i.e. a scaling of the standard inner product kernel, so $\|\boldsymbol{\beta}\|_{K_0} \propto \|\boldsymbol{\beta}\|_2$. Thus, in the kernel regime, gradient flow leads to the minimum $\ell_2$ norm solution, $\boldsymbol{\beta}_{L2}^* \doteq \arg\min_{X\boldsymbol{\beta}=y} \|\boldsymbol{\beta}\|_2$. Following Chizat and Bach (2018) and the discussion in Section 3, we thus expect that $\lim_{\alpha \to \infty} \boldsymbol{\beta}_\alpha(\infty) = \boldsymbol{\beta}_{L2}^*$, and we also show this below.

In contrast, Gunasekar et al. (2017) shows that as $\alpha \to 0$, gradient flow leads instead to the minimum $\ell_1$ norm solution $\lim_{\alpha \to 0} \boldsymbol{\beta}_\alpha(\infty) = \boldsymbol{\beta}_{L1}^* \doteq \arg\min_{X\boldsymbol{\beta}=y} \|\boldsymbol{\beta}\|_1$. This is the "rich regime." Comparing this with the kernel regime, we already see two very distinct behaviors and, in high dimensions, two very different inductive biases. In particular, the rich regime's bias is *not* an RKHS norm for any choice of kernel. Can we charactarize and understand the transition between the two regimes as $\alpha$ transitions from very small to very large? The following theorem does just that.

**Theorem 1.** *For any $0 < \alpha < \infty$,*

$$\beta_\alpha(\infty) = \hat{\boldsymbol{\beta}}_\alpha \doteq \arg\min_{\boldsymbol{\beta}} Q_\alpha(\boldsymbol{\beta}) \; s.t. \; X\boldsymbol{\beta} = \mathbf{y}, \tag{5}$$

*where $Q_\alpha(\boldsymbol{\beta}) = \sum_{i=1}^d q\left(\frac{\beta_i}{\alpha^2}\right)$ and $q(z) = \int_0^z \text{arcsinh}\left(\frac{u}{2}\right) du = 2 - \sqrt{4 + z^2} + z\,\text{arcsinh}\left(\frac{z}{2}\right)$*

**Proof sketch** The proof in Appendix A proceeds by showing the gradient flow dynamics on $\mathbf{w}$ lead to a solution of the form

$$\boldsymbol{\beta}_\alpha(\infty) = \alpha^2 \left( \exp\left( -4X^\top \int_0^\infty \boldsymbol{r}_\alpha(t) dt \right) - \exp\left( 4X^\top \int_0^\infty \boldsymbol{r}_\alpha(t) dt \right) \right) \tag{6}$$

where $\boldsymbol{r}_\alpha(t) = X\boldsymbol{\beta}_\alpha(t) - \mathbf{y}$. While evaluating the integral would be very difficult, the fact that

$$\boldsymbol{\beta}_\alpha(\infty) \in \left\{ \alpha^2 \left( \exp\left( -X^\top \bar{\boldsymbol{r}} \right) - \exp\left( X^\top \bar{\boldsymbol{r}} \right) : \bar{\boldsymbol{r}} \in \mathbb{R}^N \right) \right\} \tag{7}$$

already provides a dual certificate for the KKT conditions for $\min_{\boldsymbol{\beta}} Q_\alpha(\boldsymbol{\beta})$ s.t. $X\boldsymbol{\beta} = \mathbf{y}$. □

In light of Theorem 1, the function $Q_\alpha$ (referred to elsewhere as the "hypentropy" function (Ghai et al., 2019)) can be understood as an implicit regularizer which biases the gradient flow solution towards one particular zero-error solution out of the many possibilities. As $\alpha$ ranges from 0 to $\infty$, the $Q_\alpha$ regularizer interpolates between the $\ell_1$ and $\ell_2$ norms, as illustrated (labelled $D = 2$) in Figure 2a, which shows the coordinate function $q$. As $\alpha \to \infty$ we have that $\boldsymbol{\beta}_i/\alpha^2 \to 0$, and so the behaviour of $Q_\alpha(\boldsymbol{\beta})$ is controlled by the behaviour of $q(z)$ around $z = 0$. In this regime

---

[2]Our results can be generalized to non-uniform initialization, "biased initiliation" (i.e. where $\mathbf{w}_- \neq \mathbf{w}_+$ at initialization), or the asymmetric parameterization $f((u, v), \mathbf{x}) = \sum_i u_i v_i x_i$, however this complicates the presentation without adding much insight.

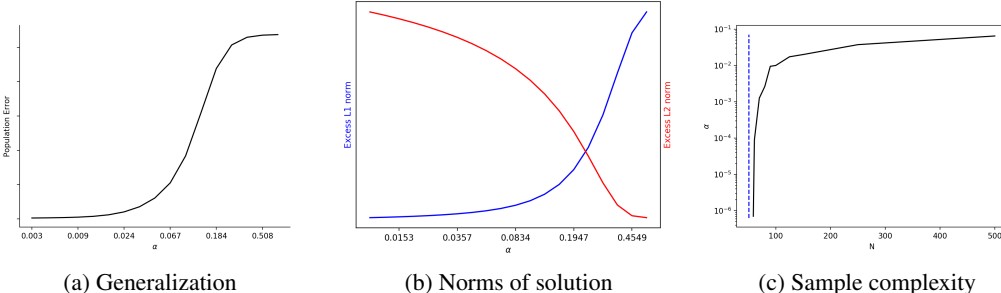

(a) Generalization          (b) Norms of solution          (c) Sample complexity

Figure 1: In (a), the population error of the gradient flow solution vs. $\alpha$ in the sparse regression problem described in Section 4. In (b), the excess $\ell_1$ norm (blue) and excess $\ell_2$ norm (red) of the gradient flow solution, i.e. $\|\boldsymbol{\beta}_\alpha(\infty)\|_1 - \|\boldsymbol{\beta}_{L1}^*\|_1$ and $\|\boldsymbol{\beta}_\alpha(\infty)\|_2 - \|\boldsymbol{\beta}_{L2}^*\|_2$. In (c), the largest $\alpha$ such that $\boldsymbol{\beta}_\alpha(\infty)$ achieves population error at most $0.025$ is shown. The dashed line indicates the number of samples needed by $\boldsymbol{\beta}_{L1}^*$.

$q(z) = \frac{z^2}{4} + O(z^4)$ is quadratic, and so $Q_\alpha(\beta) \tilde{\propto} \sum_i \beta_i^2 = \|\boldsymbol{\beta}\|_2^2$. On the other hand when $\alpha \to 0$, $\beta_i/\alpha^2 \to \infty$ is governed by the asymptotic behaviour $q(z) = \Theta(z \log z)$ as $z \to \infty$. In this regime $Q_\alpha(\boldsymbol{\beta}) \tilde{\propto} \sum_i \frac{\beta_i}{\alpha^2} \log \frac{\beta_i}{\alpha^2} \propto \|\boldsymbol{\beta}\|_1 + o(1)$. For any initialization scale $\alpha$, the function $Q_\alpha$ describes exactly how training will interpolate between the kernel and rich regimes. The following Theorems, proven in Appendix B, provide a quantitative statement of how the $\ell_1$ and $\ell_2$ norms are approached as $\alpha \to 0$ and $\alpha \to \infty$ respectively:

**Theorem 2.** *For any* $0 < \epsilon < d$

$$\alpha \leq \min\left\{ \left(2(1+\epsilon)\|\boldsymbol{\beta}_{L1}^*\|_1\right)^{-\frac{2+\epsilon}{2\epsilon}}, \exp\left(-d/(\epsilon\|\boldsymbol{\beta}_{L1}^*\|_1)\right) \right\} \implies \|\hat{\boldsymbol{\beta}}_\alpha\|_1 \leq (1+\epsilon)\|\boldsymbol{\beta}_{L1}^*\|_1$$

$$\alpha \geq \sqrt{2(1+\epsilon)(1+2/\epsilon)\|\boldsymbol{\beta}_{L2}^*\|_2} \implies \|\hat{\boldsymbol{\beta}}_\alpha\|_2^2 \leq (1+\epsilon)\|\boldsymbol{\beta}_{L2}^*\|_2^2$$

Theorem 2 indicates a certain asymmetry between reaching the rich and kernel regimes: polynomially large $\alpha$ suffices to approximate $\boldsymbol{\beta}_{L2}^*$ to a very high degree of accuracy. On the other hand, *exponentially* small $\alpha$ is sufficient to approximate $\boldsymbol{\beta}_{L1}^*$, and Lemma 2 in Appendix B proves that $\alpha \leq d^{-\Omega(1/\epsilon)}$ is *necessary* in order for $Q_\alpha$ to be proportional to the $\ell_1$ norm, which indicates that $\alpha$ must be exceedingly small to approximate $\boldsymbol{\beta}_{L1}^*$ for certain problems.

This suggestive of an explanation for the difficulty of demonstrating rich regime behavior empirically in matrix factorization problems (Gunasekar et al., 2017; Arora et al., 2019a). If the initialization really needs to be exponentially small, then conducting experiments in this regime may be infeasible for practical reasons.

In order to understand the effects of initialization on generalization, consider a simple sparse regression problem, where $\mathbf{x}^{(1)}, \dots, \mathbf{x}^{(N)} \sim N(0, I)$ and $y^{(n)} = \langle \boldsymbol{\beta}^*, \mathbf{x}^{(n)} \rangle + N(0, 0.01)$ where $\boldsymbol{\beta}^*$ is $r^*$-sparse and its non-zero entries are $1/\sqrt{r^*}$. When $N \leq d$, gradient flow will reach a zero training error solution, however, not all of these solutions will generalize the same. With $N = \Theta(r^* \log d)$ samples, the rich regime, i.e. the minimum $\ell_1$ norm solution will generalize well. However, even though we can fit the training data perfectly well, we should not expect any generalization in the kernel regime with this sample size ($N = \Omega(d)$ samples would be needed that regime), see Figure 1c. In this case, to generalize well may require using very small initialization, and generalization will improve as we decrease $\alpha$. From an optimization perspective this is unfortunate because $\mathbf{w} = 0$ is a saddle point, so taking $\alpha \to 0$ drastically increases the time needed to escape the saddle point.

Thus, there is a tension here between generalization and optimization: a smaller $\alpha$ might improve generalization, but it makes optimization trickier. This suggests that in practice we would want to compromise, and operate just at the edge of the rich regime, using the largest $\alpha$ that still allows for generalization. This is borne out in our neural network experiments in Section 7, where standard initialization schemes correspond to being right on the edge of entering the kernel regime, where we expect models to both generalize well and avoid serious optimization difficulties.

The tension between optimization and generalization can also be seen through a tradeoff between the sample size and the largest $\alpha$ we can use and still generalize. In Figure 1c, for each sample size $N$, we plot the largest $\alpha$ for which the gradient flow solution $\hat{\beta}_\alpha$ achieves population risk below

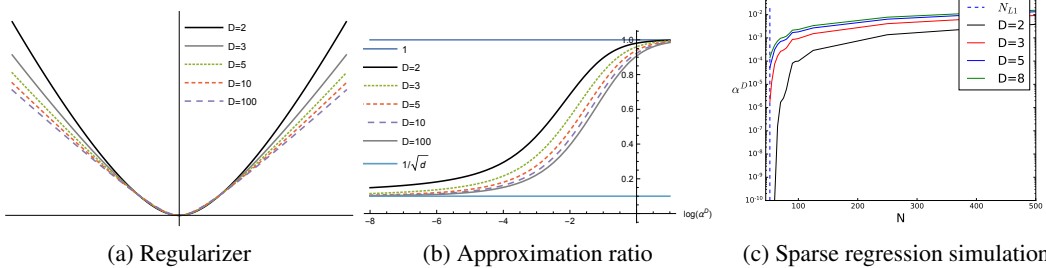

| (a) Regularizer | (b) Approximation ratio | (c) Sparse regression simulation |

Figure 2: (a) $q_D(z)$ for several values of $D$. (b) The ratio $\frac{Q_\alpha^D(e_1)}{Q_\alpha^D(\mathbf{1}_d/\|\mathbf{1}_d\|_2)}$ as a function of $\alpha$, where $e_1 = [1, 0, 0, \ldots, 0]$ is the first standard basis vector and $\mathbf{1}_d = [1, 1, \ldots, 1]$ is the all ones vector in $\mathbb{R}^d$. This captures the transition between approximating the $\ell_2$ norm (where the ratio is 1) and the $\ell_1$ norm (where the ratio is $1/\sqrt{d}$). (c) sparse regression simulation as in Figure 1, using different order models. The y-axis is $\alpha^D$ (the scale of $\boldsymbol{\beta}$ at initialization) needed to recover the planted predictor to accuracy 0.025. The dashed line indicates the number of samples needed in order for $\boldsymbol{\beta}_{L1}^*$ to approximate the plant.

some threshold. As $N$ approaches the number of samples needed for $\boldsymbol{\beta}_{L1}^*$ to generalize (the vertical dashed line), $\alpha$ must become extremely small. However, generalization is much easier when the number of samples is only slightly larger, and we can use much more moderate initialization.

## 5 HIGHER ORDER MODELS

In the previous Section, we considered a 2-homogeneous model, corresponding to a simple depth-2 "diagonal" network. Deeper models correspond to higher order homogeneity (a depth-$D$ ReLU or linear network is $D$-homogeneous), motivating us to understand the effect of the order of homogeneity on the transition between the regimes. We therefore generalize our model and consider:

$$F_D(\mathbf{w}) = \boldsymbol{\beta}_{\mathbf{w},D} = \mathbf{w}_+^D - \mathbf{w}_-^D \quad \text{and} \quad f_D(\mathbf{w}, \mathbf{x}) = \langle \mathbf{w}_+^D - \mathbf{w}_-^D, \mathbf{x} \rangle \tag{8}$$

We again consider initializing all weights equally so $\mathbf{w}_0 = \mathbf{1}$. As before, this is just a linear regression model with an unconventional parametrization. It is equivalent to a depth-$D$ matrix factorization model with commutative measurement matrices, as studied by Arora et al. (2019a), and can be thought of as a depth-$D$ diagonal linear network.

We can again study the effect of the scale of the initialization $\alpha$ on the implicit bias. Let $\boldsymbol{\beta}_{\alpha,D}(\infty)$ denote the limit of gradient flow on $\mathbf{w}$ when initialized at $\alpha\mathbf{1}$ for the $D$-homogeneous model. Using the same approach as in Section 4, in Appendix C we show:

**Theorem 3.** *For any $\alpha$ and $D \geq 3$, if gradient flow reaches a solution $X\boldsymbol{\beta}_{\alpha,D}(\infty) = y$, then*

$$\boldsymbol{\beta}_{\alpha,D}(\infty) = \arg\min_{\boldsymbol{\beta}} Q_\alpha^D(\boldsymbol{\beta}) \text{ s.t. } \mathbf{X}\boldsymbol{\beta} = \mathbf{y}$$

*where $Q_\alpha^D(\boldsymbol{\beta}) = \sum_{i=1}^d q_D(\beta_i/\alpha^D)$ and $q_D = \int h_D^{-1}$ is the antiderivative of the unique inverse of $h_D(z) = (1 - z)^{-\frac{D}{D-2}} - (1 + z)^{-\frac{D}{D-2}}$ on $[-1, 1]$. Furthermore, $\lim_{\alpha \to 0} \boldsymbol{\beta}_{\alpha,D}(\infty) = \boldsymbol{\beta}_{L1}^*$ and $\lim_{\alpha \to \infty} \boldsymbol{\beta}_{\alpha,D}(\infty) = \boldsymbol{\beta}_{L2}^*$.*

In the two extremes we see that we again get the minimum $\ell_2$ solution in the kernel regime, and more interestingly, for any depth $D \geq 2$, we get the same minimum $\ell_1$ norm solution in the rich regime, as has also been observed by Arora et al. (2019a). The fact that the rich regime solution does not change with depth is perhaps surprising, and does not agree from what is obtained with explicit regularization (regularizing $\|w\|_2$ is equivalent to $\|\beta\|_{2/D}$ regularization), nor with implicit regularization on logistic-type loss Gunasekar et al. (2017).

Although the two extremes do not change as we go beyond $D = 2$, what does change is the intermediate regime (the regularizer $Q_\alpha^D$ is unique and cannot be obtained with any other order $D' \neq D$), as well as the sharpness of the transition into the extreme regimes, as illustrated in Figures 2a-2c. The most striking difference is that for order $D > 2$ the scale of $\alpha$ needed to approximate the $\ell_1$ is polynomial rather then exponential, yielding a much quicker transition to the "rich regime", and

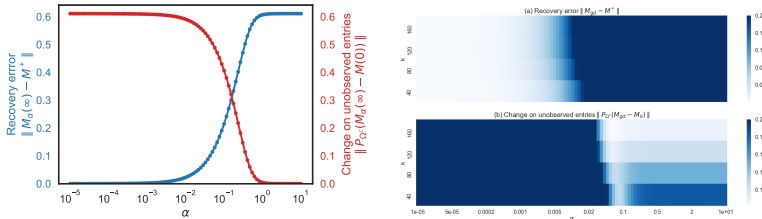

Figure 3: **Regimes in Matrix Completion** We generated a $10 \times 10$ rank-one matrix completion problem with ground truth $M^* = u^*(v^*)^\top$ by generating $u^*, v^* \in \mathbb{R}^{10}$ with i.i.d. $\mathcal{N}(0,1)$ entries and observing $N = 60$ random entries $\Omega$. We fit the observed entries by minimizing the squared loss on a matrix factorization model $F(U, V) = UV^\top$ with $U, V \in \mathbb{R}^{d \times 2k}$. For different scalings $\alpha$, we examine the matrix $M(\infty)$ reached by gradient flow on $U, V$ (solved using python ODE solvers) and plot (i) the reconstruction error on unobserved entries $\sum_{ij \notin \Omega}(M_{ij}^* - M(\infty)_{ij})^2$, and (ii) the amount by which the unobserved entries changed during optimization $\sum_{ij \notin \Omega}(M(\infty)_{ij} - M(0)_{ij})^2$. In (a) we used $k = 2d$ and initialized to $U_0 = V_0 = \alpha I$. In (b) for varying $k$, we initialized to $U_0 = \alpha \bar{U}_0$ and $V_0 = \alpha \bar{V}_0$ with $\bar{U}_0, \bar{V}_0 \in \mathbb{R}^{d \times k}$ with i.i.d. $\mathcal{N}(0,1)$ entries. For large $k$, the tangent kernel converges to the kernel corresponding to the Frobenius norm.

allowing near-optimal sparse regression with reasonable initialization scales. Increasing $D$ further hastens the transition. This might also help explain some of the empirical observations about the benefit of depth in deep matrix factorization Arora et al. (2019a).

## 6 DEMONSTRATION IN MATRIX COMPLETION

We now turn to a more complex depth two model, namely a matrix factorization model, and demonstrate similar transitions empirically. Specifically, we consider the model over matrix inputs $X \in \mathbb{R}^{d \times d}$ defined by $f((U, V), X) = \langle UV^\top, X \rangle$, where $U, V \in \mathbb{R}^{d \times k}$. This corresponds to linear predictors over matrix arguments specified by $F(U, V) = UV^\top$. In overparameterized regime with $k \geq d$, the parameterization itself does not introduce any explicit rank constraints. We consider here a random low rank matrix completion problem where $X_n = e_{i_n} e_{j_n}^\top$ represents an uniform random observation of entry $(i_n, j_n)$ of a planted low rank matrix $M^*$ of rank $r^* \ll d$: $y_n = \langle M^*, X_n \rangle = M_{ij}^*$. For underdetermined problems where $N \ll d^2$, there are many trivial global minimizers, most of which are not low rank and hence will not guarantee recovery. As was demonstrated empirically by Gunasekar et al. (2017) and also proven rigorously for Gaussian measurements by Li et al. (2018), as $\alpha \to 0$ gradient flow implicitly regularizes the nuclear norm, which for random measurements leads to recovery of ground truth (Candès and Recht, 2009; Recht et al., 2010): these are very different and rich implicit biases that are not RKHS norms.

Crucially, the reconstruction results in Gunasekar et al. (2017) and Li et al. (2018) are dependent on initialization with scale $\alpha \to 0$. Here we further explore the role of initialization. Similar to Section 4, in order to get unbiased 0 initialization, we consider $k \geq 2d$ and initialization of the form $U(0) = \alpha [U_0, -U_0]$ and $V(0) = \alpha [V_0, V_0]$, where $U_0, V_0 \in \mathbb{R}^{d \times k/2}$. We will study implicit bias of gradient flow over the factorized parameterization with above initialization.

For matrix completion problems with $X = e_{i_X} e_{j_X}^\top$, the tangent kernel at initialization is given by $K_0(X, X') \propto \langle U_0[i_X, :], U_0[i_{X'}, :] \rangle \mathbf{1}(j_X = j_{X'}) + \langle V_0[j_X, :], V_0[j_{X'}, :] \rangle \mathbf{1}(i_X = i_{X'})$. This defaults to the trivial delta kernel $K(X, X') = \mathbf{1}(i_X = i_{X'}) \cdot \mathbf{1}(j_X = j_{X'})$ for the two special cases (a) $U_0, V_0$ have orthogonal columns (e.g. $U_0 = V_0 = I$), or (b) $U_0, V_0$ have independent Gaussian entries and $k \to \infty$. In these cases, minimizing the RKHS norm of the tangent kernel corresponds to returning a zero imputed matrix (minimum Frobenius norm solution). Figure 3 demonstrates the behaviour of gradient flow updates in the "rich" regime (where for $\alpha \to 0$ recovers the ground truth) and in the "kernel" regime (where for large $\alpha$, there are no updates to the unobserved entries).

## 7 NEURAL NETWORK EXPERIMENTS

In the preceding sections, we intentionally focused on the simplest possible models in which a kernel-to-rich transition can be observed, in order to isolate this phenomena and understand it in

detail. In those simple models, we were able to obtain a complete analytic description of the transition. Obtaining such a precise description in more complex models is somewhat optimistic at this point, as we do not yet have a satisfying description of even just the rich regime. Instead, we now provide empirical evidence suggesting that also for non-linear and realistic networks, the scale of initialization induces a transition into and out of a "kernel" regime, and that to reach good generalization we must operate outside of the "kernel" regime. To track whether we are in the "kernel" regime, we track how much the gradient $\nabla_{\mathbf{w}} f(\mathbf{w}(t), \mathbf{x})$ changes throughout training. In particular, we define the *gradient distance* to be the cosine distance between the initial tangent kernel feature map $\nabla_{\mathbf{w}} f(\mathbf{w}(0), \mathbf{x})$ and the final tangent kernel feature map $\nabla_{\mathbf{w}} f(\mathbf{w}(T), \mathbf{x})$.

In Figures 4a and 4b, we see that also for a non-linear ReLU network, we remain in the kernel regime when the initialization is large, and that exiting from the kernel regime is necessary in order to achieve small test error on the synthetic data. Interestingly, when $\alpha^D \approx 1$, the models achieve good test error but have smaller gradient distance which, not coincidentally, corresponds to using the out-of-the-box Uniform He initialization. This lies on the boundary between the rich and kernel regimes, which is desirable due to the learning vs. optimization tradeoffs discussed in Section 4. On MNIST data, Figure 4e shows that previously published successes with training overly wide depth-2 ReLU networks without explicit regularization (e.g. Neyshabur et al., 2014) relies on the initialization being small, i.e. being outside of the "kernel regime". In fact, the 2.4% test error reached for large initialization is no better than what can be achieved with a linear model over a random feature map. Turning to a more realistic network, 4f shows similar behavior when training a VGG11-like network on CIFAR10.

So far, we attempted to use the best fixed stepsize for each initialization (i.e. achieving the best test error). But as demonstrated in Figures 4c and 4d, the stepsize choice can also have a significant effect, with larger stepsizes allowing one to exit the kernel regime even at an initialization scale where a smaller stepsize would remain trapped in the kernel regime. Further analytic and empirical studies are necessary in order to understand the joint behavior of the stepsize and initialization scale.

# 8 DISCUSSION

The main point of this paper is to emphasize the distinction between the "kernel" regime in training overparametrized multi-layered networks, and the "rich" (active, adaptive) regime, show how the scaling of the initialization can transition between them, and understand this transition in detail. We argue that rich inductive bias that enables generalization may arise in the rich regime, but that focusing on the kernel regime restricts us to only what can be done with an RKHS. By studying the transition we also see a tension between generalization and optimization, which suggests we would tend to operate just on the edge of the rich regime, and so understanding this transition, rather then just the extremes, is important. Furthermore, we see that at the edge of the rich regime, the implicit bias of gradient descent differs substantively from that of explicit regularization. Although in our theoretical study we focused on a simple model so that we can carry out a complete and exact analysis analytically, our experiments show that this is representative of the behaviour also in other homogeneous models, and serve as a basis of a more general understanding.

**Effect of Width**  Our treatment focused on the effect of scale on the transition between the regimes, and we saw that, as pointed out by Chizat and Bach, we can observe a very meaningful transition between a kernel and rich regime even for finite width parametric models. The transition becomes even more interesting if the width of the model (the number of units per layer, and so also the number of parameters) increases towards infinity. In this case, we must be careful as to how the initialization of each individual unit scales when the total number of units increase, and which regime we fall in to is controlled by the relative scaling of the width and the scale of individual units at initialization. This is demonstrated, for example, in Figure 5, which shows the regime change in matrix factorization problems, from minimum Frobenius norm recovery (the kernel regime) to minimum nuclear norm recovery (the rich regime), as a function of both the number of factors $k$ and the scale of initialization of each factor $\alpha$ . As is expected, the scale $\alpha$ at which we see the transition decreases as the model becomes wider, but further study is necessary to obtain a complete understanding of this scaling.

A particularly interesting aspect of infinite width networks is that, unlike for fixed-width networks, it may be possible to scale $\alpha$ relative to the width $k$ such that at the infinite-width limit we would

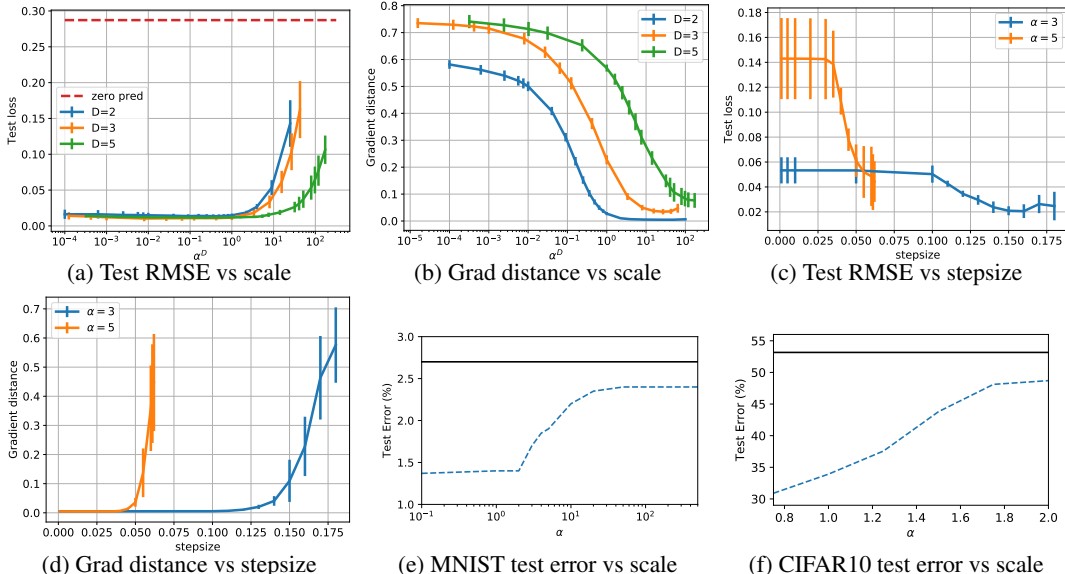

(a) Test RMSE vs scale  (b) Grad distance vs scale  (c) Test RMSE vs stepsize

(d) Grad distance vs stepsize  (e) MNIST test error vs scale  (f) CIFAR10 test error vs scale

Figure 4: **Synthetic Data**: we generated a small regression training set in $\mathbb{R}^2$ by sampling 10 points uniformly from the unit circle, and labelling them with a 1 hidden layer teacher network with 3 hidden units. We trained overparametrized depth-$D$, ReLU networks with 30 units per layer with squared loss using full GD and a small stepsize 0.01. The weights of the network are set using the Uniform He initialization, and then multiplied by $\alpha$. The model is trained until $\approx 0$ training loss. Shown in (a) and (b) are the test error and grad distance vs. the depth-adjusted scale of the initialization, $\alpha^D$, when a small constant stepsize is used. for (c) and (d), we fix $\alpha$ near the transition into the kernel regime, and show the test error and grad distance vs. the stepsize. **MNIST**: we trained a depth-2 with 5000 hidden units with cross-entropy loss using SGD until it reached 100% training accuracy. The stepsizes were optimally tuned for each $\alpha$ individually. In (e), the dashed line shows the test error of the resulting network vs. $\alpha$. We repeated the experiment, but froze the bottom layer and only trained the output layer until convergence. The solid line shows the test error of this predictor vs $\alpha$. **CIFAR10**: we trained a VGG11-like deep convolutional network with cross-entropy loss using SGD and a small stepsize $10^{-4}$ for 2000 epochs; all models reached 100% training accuracy. In (f), the dashed line shows the final test error vs. $\alpha$. We repeated the experiment freezing the bottom 10 layers and training only the output layer–the solid line shows this model's test error. See Appendix E for full details about all of the experiments.

have an (asymptotically) unbiased predictor at initialization $\lim_{k\to\infty} F_k(\mathbf{w}(0)) = 0$, or at least a non-exploding initialization $\limsup_{k\to\infty} \|F_k(\mathbf{w}(0))\| = O(1)$, even with random initialization (without a doubling trick leading to artificially unbiased initialization), while still being in the kernel regime. For two-layer networks with ReLU activation, Arora et al. (2019b) showed that with width $k \geq \text{poly}(1/\max_{\|x\|_2 \leq 1, N} \|f_0(x)\|)$ the gradient dynamics stay in the kernel regime forever.

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

## A    PROOF OF THEOREM 1

It is straightforward, given the expression for $Q_\alpha$, to prove that $\hat{\boldsymbol{\beta}}_\alpha$ is the minimum $Q_\alpha$ solution to $X\boldsymbol{\beta} = \mathbf{y}$. In other words, if we had been able to guess from the beginning that the implicit bias would be governed by $Q_\alpha$, it would have been easy to prove this fact. However, a key contribution of this work is in developing a method for determining what the implicit bias is *when we do not already have a good guess*.

We will first describe a general approach for deriving the implied regularizer reached at the limit of some gradient flow dynamics which minimize the loss. In particular, our approach will apply whenever solutions of the gradient flow dynamics can be written as $\boldsymbol{\beta}_\alpha(t) = f_\alpha(X^\top \nu(t))$ for some $\nu(t)$.

The first step is to analyze the gradient flow dynamics of $\boldsymbol{\beta}_\alpha$ and show that

$$\begin{aligned} X\boldsymbol{\beta}_\alpha(\infty) &= \mathbf{y} \\ \boldsymbol{\beta}_\alpha(\infty) &= f_\alpha(X^\top \nu(\infty)) \end{aligned} \tag{9}$$

for some function $f_\alpha$ and some vector $\nu(\infty)$. It is not important for our approach to know exactly what $\nu(\infty)$ is. This is useful because calculating this $\nu(\infty)$ will often be difficult, even for the simple examples we consider.

The next step is to suppose that there is some function $Q_\alpha$ such that

$$\boldsymbol{\beta}_\alpha(\infty) = \arg\min_{\boldsymbol{\beta}} Q_\alpha(\boldsymbol{\beta}) \quad \text{s.t.} \quad X\boldsymbol{\beta} = \mathbf{y} \tag{10}$$

and to write down the KKT optimality conditions for (10):

$$\begin{aligned} X\boldsymbol{\beta}^* &= \mathbf{y} \\ \exists \nu \ \nabla Q_\alpha(\boldsymbol{\beta}^*) &= X^\top \nu \end{aligned} \tag{11}$$

Finally, we connect (9) with (11). Specifically, $\boldsymbol{\beta}_\alpha(\infty)$ satisfies the first KKT condition, and, if $\boldsymbol{\beta}_\alpha(\infty) = \beta^*$ then by taking $\nu = \nu(\infty)$ we have

$$\nabla Q_\alpha(\boldsymbol{\beta}_\alpha(\infty)) = \nabla Q_\alpha(f_\alpha(X^\top \nu)) = X^\top \nu \tag{12}$$

It follows that $\nabla Q_\alpha(\boldsymbol{\beta}) = f_\alpha^{-1}(\boldsymbol{\beta})$. Therefore, to derive $Q_\alpha$ we simply invert $f_\alpha$ and integrate its inverse. We prove Theorems 1 and 3 using this method.

**Theorem 1.** *For any $0 < \alpha < \infty$,*

$$\beta_\alpha(\infty) = \hat{\boldsymbol{\beta}}_\alpha \doteq \arg\min_{\boldsymbol{\beta}} Q_\alpha(\boldsymbol{\beta}) \ \text{s.t.} \ X\boldsymbol{\beta} = \mathbf{y}, \tag{5}$$

*where $Q_\alpha(\boldsymbol{\beta}) = \sum_{i=1}^d q\left(\frac{\beta_i}{\alpha^2}\right)$ and $q(z) = \int_0^z \operatorname{arcsinh}\left(\frac{u}{2}\right) du = 2 - \sqrt{4 + z^2} + z \operatorname{arcsinh}\left(\frac{z}{2}\right)$*

*Proof.* We begin by calculating the gradient flow dynamics on $\mathbf{w}$, since the linear predictor $\boldsymbol{\beta}_\alpha(\infty)$ is given by $F$ applied to the limit of the gradient flow dynamics on $\mathbf{w}$. Recalling that $\tilde{X} = [X \quad -X]$,

$$\dot{\mathbf{w}}_\alpha(t) = -\nabla L(\mathbf{w}_\alpha(t)) = -\nabla\left(\left\|\tilde{X}\mathbf{w}_\alpha(t)^2 - y\right\|_2^2\right) = -2\tilde{X}^\top r_\alpha(t) \circ \mathbf{w}_\alpha(t) \tag{13}$$

where the residual $r_\alpha(t) \triangleq \tilde{X}\mathbf{w}_\alpha(t)^2 - y$, and $a \circ b$ denotes the element-wise product of $a$ and $b$. It is easily confirmed that these dynamics have a solution:

$$\mathbf{w}_\alpha(t) = \mathbf{w}_\alpha(0) \circ \exp\left(-2\tilde{X}^\top \int_0^t r_\alpha(s)ds\right) \tag{14}$$

This immediately gives an expression for $\boldsymbol{\beta}_\alpha(t)$:

$$\boldsymbol{\beta}_\alpha(t) = \mathbf{w}_{\alpha,+}(t)^2 - \mathbf{w}_{\alpha,-}(t)^2 \tag{15}$$

$$= \alpha^2 \left( \exp\left(-4X^\top \int_0^t r_\alpha(s)ds\right) - \exp\left(4X^\top \int_0^t r_\alpha(s)ds\right) \right) \tag{16}$$

$$= 2\alpha^2 \sinh\left(-4X^\top \int_0^t r_\alpha(s)ds\right) \tag{17}$$

In addition, this problem satisfies the strict saddle property (Ge et al., 2015) (Zhao et al., 2019, Lemma 2.1), therefore gradient flow will converge to a zero-error solution, i.e. $X\boldsymbol{\beta}_\alpha(\infty) = y$. Thus, we conclude that $\boldsymbol{\beta}_\alpha(\infty)$ is a global minimum with zero error, i.e. $X\boldsymbol{\beta}_\alpha(\infty) = \mathbf{y}$.

Thus, we have established that

$$\begin{aligned} X\boldsymbol{\beta}_\alpha(\infty) &= \mathbf{y} \\ \boldsymbol{\beta}_\alpha(\infty) &= f_\alpha(X^\top \nu(\infty)) \end{aligned} \tag{18}$$

for $f_\alpha(z) = 2\alpha^2 \sinh(z)$ and $\nu(\infty) = -4 \int_0^t r_\alpha(s)ds$. This corresponds to (9) from our general approach detailed above. Continuing from (12) we have that

$$\nabla Q_\alpha(\boldsymbol{\beta}) = f_\alpha^{-1}(\boldsymbol{\beta}) = \operatorname{arcsinh}\left(\boldsymbol{\beta}/2\alpha^2\right) \tag{19}$$

Integrating this expression completes the proof. $\qquad\square$

## B  PROOF OF THEOREM 2

**Lemma 1.** *For any* $\beta \in \mathbb{R}^d$,

$$\alpha \leq \alpha_1\left(\epsilon, \|\beta\|_1, d\right) := \min\left\{1, \sqrt{\|\beta\|_1}, (2\|\beta\|_1)^{-\frac{1}{2\epsilon}}, \exp\left(-\frac{d}{2\epsilon\|\beta\|_1}\right)\right\}$$

*guarantees that*

$$\|\beta\|_1 (1 - \epsilon) \leq \frac{\alpha^2}{\ln(1/\alpha^2)} Q(\beta/\alpha^2) \leq \|\beta\|_1 (1 + \epsilon)$$

*Proof.* First, we show that $Q(\beta/\alpha^2) = Q_\alpha(|\beta|/\alpha^2)$. Observe that $g(x) = x\arcsin(x/2)$ is even because $x$ and $\arcsin(x/2)$ are odd. Therefore,

$$Q(\beta/\alpha^2) = \sum_{i=1}^d 2 - \sqrt{4 + \frac{\beta_i^2}{\alpha^4}} + \frac{\beta_i}{\alpha^2} \operatorname{arcsinh}\left(\frac{\beta_i}{2\alpha^2}\right) \tag{20}$$

$$= \sum_{i=1}^d 2 - \sqrt{4 + \frac{\beta_i^2}{\alpha^4}} + g\left(\frac{\beta_i}{\alpha^2}\right) \tag{21}$$

$$= \sum_{i=1}^d 2 - \sqrt{4 + \frac{|\beta_i|^2}{\alpha^4}} + g\left(\left|\frac{\beta_i}{\alpha^2}\right|\right) \tag{22}$$

$$= Q_\alpha(|\beta|/\alpha^2) \tag{23}$$

Therefore, we can rewrite

$$\frac{\alpha^2}{\ln(1/\alpha^2)}Q(\beta/\alpha^2) = \frac{\alpha^2}{\ln(1/\alpha^2)}Q(|\beta|/\alpha^2) \tag{24}$$

$$= \sum_{i=1}^{d} \frac{2\alpha^2}{\ln(1/\alpha^2)} - \frac{\sqrt{4\alpha^4 + \beta_i^2}}{\ln(1/\alpha^2)} + \frac{|\beta_i|}{\ln(1/\alpha^2)}\operatorname{arcsinh}\left(\frac{|\beta_i|}{2\alpha^2}\right) \tag{25}$$

$$= \sum_{i=1}^{d} \frac{2\alpha^2}{\ln(1/\alpha^2)} - \frac{\sqrt{4\alpha^4 + \beta_i^2}}{\ln(1/\alpha^2)} + \frac{|\beta_i|}{\ln(1/\alpha^2)}\ln\left(\frac{|\beta_i|}{2\alpha^2} + \sqrt{1 + \frac{\beta_i^2}{4\alpha^4}}\right) \tag{26}$$

$$= \sum_{i=1}^{d} \frac{2\alpha^2}{\ln(1/\alpha^2)} - \frac{\sqrt{4\alpha^4 + \beta_i^2}}{\ln(1/\alpha^2)} + |\beta_i|\left(1 + \frac{\ln\left(\frac{|\beta_i|}{2} + \sqrt{\alpha^4 + \frac{\beta_i^2}{4}}\right)}{\ln(1/\alpha^2)}\right) \tag{27}$$

Using the fact that

$$|a| \le \sqrt{a^2 + b^2} \le |a| + |b| \tag{28}$$

we can bound for $\alpha < 1$

$$\frac{\alpha^2}{\ln(1/\alpha^2)}Q(\beta/\alpha^2) \le \sum_{i=1}^{d} \frac{2\alpha^2}{\ln(1/\alpha^2)} - \frac{2\alpha^2}{\ln(1/\alpha^2)} + |\beta_i|\left(1 + \frac{\ln\left(\frac{|\beta_i|}{2} + \alpha^2 + \frac{|\beta_i|}{2}\right)}{\ln(1/\alpha^2)}\right) \tag{29}$$

$$= \sum_{i=1}^{d} |\beta_i|\left(1 + \frac{\ln\left(|\beta_i| + \alpha^2\right)}{\ln(1/\alpha^2)}\right) \tag{30}$$

$$\le \|\beta\|_1 \left(1 + \max_{i \in [d]} \frac{\ln\left(|\beta_i| + \alpha^2\right)}{\ln(1/\alpha^2)}\right) \tag{31}$$

$$\le \|\beta\|_1 \left(1 + \frac{\ln\left(\|\beta\|_1 + \alpha^2\right)}{\ln(1/\alpha^2)}\right) \tag{32}$$

$$\tag{33}$$

So, for any $\alpha \le \min\left\{1, \sqrt{\|\beta\|_1}, (2\|\beta\|_1)^{-\frac{1}{2\epsilon}}\right\}$, then

$$\frac{\alpha^2}{\ln(1/\alpha^2)}Q(\beta/\alpha^2) \le \|\beta\|_1 \left(1 + \frac{\ln\left(\|\beta\|_1 + \alpha^2\right)}{\ln(1/\alpha^2)}\right) \tag{34}$$

$$\le \|\beta\|_1 \left(1 + \frac{\ln\left(2\|\beta\|_1\right)}{\ln(1/\alpha^2)}\right) \tag{35}$$

$$\le \|\beta\|_1 (1 + \epsilon) \tag{36}$$

On the other hand, using (27) and (28) again,

$$\frac{\alpha^2}{\ln(1/\alpha^2)}Q(\beta/\alpha^2) \ge \sum_{i=1}^{d} \frac{2\alpha^2}{\ln(1/\alpha^2)} - \frac{|\beta_i| + 2\alpha^2}{\ln(1/\alpha^2)} + |\beta_i|\left(1 + \frac{\ln\left(|\beta_i|\right)}{\ln(1/\alpha^2)}\right) \tag{37}$$

$$= \sum_{i=1}^{d} |\beta_i|\left(1 + \frac{\ln\left(|\beta_i|\right) - 1}{\ln(1/\alpha^2)}\right) \tag{38}$$

Using the inequality $\ln(x) \ge 1 - \frac{1}{x}$, this can be further lower bounded by

$$\frac{\alpha^2}{\ln(1/\alpha^2)}Q(\beta/\alpha^2) \ge \sum_{i=1}^{d} |\beta_i| - \frac{1}{\ln(1/\alpha^2)} \tag{39}$$

$$= \|\beta\|_1 - \frac{d}{\ln(1/\alpha^2)} \tag{40}$$

Therefore, for any $\alpha \leq \exp\left(-\frac{d}{2\epsilon \|\beta\|_1}\right)$ then

$$\frac{\alpha^2}{\ln(1/\alpha^2)} Q(\beta/\alpha^2) \geq \|\beta\|_1 (1 - \epsilon) \tag{41}$$

We conclude that for $\alpha \leq \min\left\{1, \sqrt{\|\beta\|_1}, (2\|\beta\|_1)^{-\frac{1}{2\epsilon}}, \exp\left(-\frac{d}{2\epsilon \|\beta\|_1}\right)\right\}$ that

$$\|\beta\|_1 (1 - \epsilon) \leq \frac{\alpha^2}{\ln(1/\alpha^2)} Q(\beta/\alpha^2) \leq \|\beta\|_1 (1 + \epsilon) \tag{42}$$

$\square$

**Lemma 2.** *Fix any $\epsilon > 0$ and $d \geq \max\left\{e, 12^{4\epsilon}\right\}$. Then for any $\alpha \geq d^{-\frac{1}{4} - \frac{1}{8\epsilon}}$, $Q(\beta/\alpha^2) \not\propto \|\beta\|_1$ in the sense that there exist vectors $v, w$ such that*

$$\frac{Q\left(\frac{1}{\alpha^2} v\right)}{\|v\|_1} \geq (1 + \epsilon) \frac{Q\left(\frac{1}{\alpha^2} w\right)}{\|w\|_1}$$

*Proof.* First, recall that

$$q\left(\frac{1}{c\alpha^2}\right) = 2 - \sqrt{4 + \frac{1}{c^2\alpha^4}} + \frac{1}{c\alpha^2} \operatorname{arcsinh}\left(\frac{1}{2c\alpha^2}\right) \tag{43}$$

$$= \frac{1}{c\alpha^2}\left(2c\alpha^2 - \sqrt{4c^2\alpha^4 + 1} + \ln\left(\frac{1}{2c\alpha^2} + \sqrt{1 + \frac{1}{4c^2\alpha^4}}\right)\right) \tag{44}$$

$$= \frac{1}{c\alpha^2}\left(2c\alpha^2 - \sqrt{4c^2\alpha^4 + 1} + \ln\left(\frac{1}{c\alpha^2}\right) + \ln\left(\frac{1}{2} + \sqrt{c^2\alpha^4 + \frac{1}{2}}\right)\right) \tag{45}$$

Thus,

$$-1 + \ln\left(\frac{1}{c\alpha^2}\right) \leq c\alpha^2 q\left(\frac{1}{c\alpha^2}\right) \leq 3c\alpha^2 - 1 + \ln\left(\frac{1}{c\alpha^2}\right) \tag{46}$$

Now, consider the ratio

$$\frac{Q\left(\frac{1}{\alpha^2} e_1\right)}{Q\left(\frac{1}{\alpha^2} \frac{\mathbf{1}_d}{\|\mathbf{1}_d\|_2}\right)} = \frac{q\left(\frac{1}{\alpha^2}\right)}{dq\left(\frac{1}{\alpha^2\sqrt{d}}\right)} \tag{47}$$

$$= \frac{1}{\sqrt{d}} \frac{\alpha^2 q\left(\frac{1}{\alpha^2}\right)}{\alpha^2\sqrt{d} q\left(\frac{1}{\alpha^2\sqrt{d}}\right)} \tag{48}$$

Using (46), we conclude

$$\sqrt{d} \frac{Q\left(\frac{1}{\alpha^2} e_1\right)}{Q\left(\frac{1}{\alpha^2} \frac{\mathbf{1}_d}{\|\mathbf{1}_d\|_2}\right)} \geq \frac{-1 + \ln\left(\frac{1}{\alpha^2}\right)}{3\sqrt{d}\alpha^2 - 1 + \ln\left(\frac{1}{\alpha^2\sqrt{d}}\right)} \tag{49}$$

$$= \frac{-1 + \ln\left(\frac{1}{\alpha^2}\right)}{3\sqrt{d}\alpha^2 - 1 + \ln\left(\frac{1}{\alpha^2}\right) - \frac{1}{2}\ln(d)} \tag{50}$$

$$= 1 + \frac{\ln(d) - 6\sqrt{d}\alpha^2}{6\sqrt{d}\alpha^2 - 2 + 2\ln\left(\frac{1}{\alpha^2\sqrt{d}}\right)} \tag{51}$$

Fix any $\epsilon > 0$ and $d \geq \max\left\{e, 12^{4\epsilon}\right\}$, and set $\alpha = d^{-\frac{1}{4} - \frac{1}{8\epsilon}}$. Then

$$2d^{\frac{1}{4\epsilon}} \geq 6 \quad \text{and} \quad \frac{d^{\frac{1}{4\epsilon}}}{2} \ln d \geq 6 \tag{52}$$

$$\implies 2d^{\frac{1}{4\epsilon}} - 6 + \frac{d^{\frac{1}{4\epsilon}}}{2\epsilon} \ln d - \frac{6}{\epsilon} \geq 0 \tag{53}$$

$$\implies \left(\frac{1}{\epsilon} - \frac{1}{2\epsilon}\right) \ln d - \frac{6}{\epsilon} d^{-\frac{1}{4\epsilon}} \geq 6d^{-\frac{1}{4\epsilon}} - 2 \tag{54}$$

$$\implies \frac{1}{\epsilon} \ln d - \frac{6}{\epsilon} d^{-\frac{1}{4\epsilon}} \geq 6d^{-\frac{1}{4\epsilon}} - 2 + \frac{1}{2\epsilon} \ln d \tag{55}$$

$$\implies \ln(d) - 6\alpha^2 \sqrt{d} \geq \epsilon \left(6\alpha^2 \sqrt{d} - 2 + 2\ln\left(\frac{1}{\alpha^2 \sqrt{d}}\right)\right) \tag{56}$$

This implies that the second term of (51) is at least $\epsilon$. We conclude that for any $\epsilon > 0$ and $d \geq \max\left\{e, 12^{4\epsilon}\right\}$, $\alpha = d^{-\frac{1}{4} - \frac{1}{8\epsilon}}$ implies that

$$\frac{Q\left(\frac{1}{\alpha^2} e_1\right)}{Q\left(\frac{1}{\alpha^2} \frac{\mathbf{1}_d}{\|\mathbf{1}_d\|_2}\right)} \geq (1 + \epsilon) \frac{\|e_1\|_1}{\left\|\frac{\mathbf{1}_d}{\|\mathbf{1}_d\|_2}\right\|_1} \tag{57}$$

Consequently, for at least one of these two vectors, $Q$ is not proportional to the $\ell_1$ norm up to accuracy $O(\epsilon)$ for this value of $\alpha$.

It is straightforward to confirm that

$$\frac{d}{d\alpha} \frac{Q\left(\frac{1}{\alpha^2} e_1\right)}{Q\left(\frac{1}{\alpha^2} \frac{\mathbf{1}_d}{\|\mathbf{1}_d\|_2}\right)} \geq 0 \tag{58}$$

which concludes the proof. $\qquad \square$

**Lemma 3.** *For any $\beta \in \mathbb{R}^d$,*

$$\alpha \geq \alpha_2(\epsilon, \|w\|_2) := \sqrt{\|\beta\|_2} \left(1 + \epsilon^{-\frac{1}{4}}\right)$$

*guarantees that*

$$(1 - \epsilon) \|\beta\|_2^2 \leq 4\alpha^4 Q(\beta/\alpha^2) \leq (1 + \epsilon) \|\beta\|_2^2$$

*Proof.* The regularizer $Q$ can be written

$$Q(\beta/\alpha^2) = \sum_{i=1}^{d} \int_0^{\beta_i/\alpha^2} \operatorname{arcsinh}\left(\frac{t}{2}\right) dt \tag{59}$$

Let $\phi(z) = \int_0^{z/\alpha^2} \operatorname{arcsinh}\left(\frac{t}{2}\right) dt$, then

$$\phi(0) = 0 \tag{60}$$

$$\phi'(0) = \frac{1}{\alpha^2} \operatorname{arcsinh}\left(\frac{z}{2\alpha^2}\right)\bigg|_{z=0} = 0 \tag{61}$$

$$\phi''(0) = \frac{1}{\alpha^4 \sqrt{4 + \frac{z^2}{\alpha^4}}}\bigg|_{z=0} = \frac{1}{2\alpha^4} \tag{62}$$

$$\phi'''(0) = \frac{-z}{\alpha^8 \left(4 + \frac{z^2}{\alpha^4}\right)^{3/2}}\bigg|_{z=0} = 0 \tag{63}$$

$$\phi''''(z) = \frac{3z^2}{\alpha^{12} \left(4 + \frac{z^2}{\alpha^4}\right)^{5/2}} - \frac{1}{\alpha^8 \left(4 + \frac{z^2}{\alpha^4}\right)^{3/2}} \tag{64}$$

Also, note that

$$|\phi''''(z)| = \frac{\left|2z^2 - 4\alpha^4\right|}{\alpha^{12}\left(4 + \frac{z^2}{\alpha^4}\right)^{5/2}} \tag{65}$$

$$\leq \frac{z^2 + 2\alpha^4}{16\alpha^{12}} \tag{66}$$

Therefore, by Taylor's theorem, for some $\xi$ with $|\xi| \leq |z|$

$$\left|\phi(z) - \frac{z^2}{4\alpha^4}\right| = \frac{\phi''''(\xi)}{4!}z^4 \tag{67}$$

$$\implies \left|\phi(z) - \frac{z^2}{4\alpha^4}\right| \leq \sup_{|\xi| \leq |z|} \frac{\phi''''(\xi)}{4!}z^4 \leq \frac{z^6 + 2\alpha^4 z^4}{384\alpha^{12}} = \frac{z^2}{4\alpha^4}\frac{z^4 + 2\alpha^4 z^2}{96\alpha^8} \tag{68}$$

Therefore, for any $\beta \in \mathbb{R}^d$

$$\left|4\alpha^4 Q_\alpha(\beta) - \|\beta\|_2^2\right| = 4\alpha^4 \left|\sum_{i=1}^{d} \phi(\beta_i) - \frac{\beta_i^2}{4\alpha^4}\right| \tag{69}$$

$$\leq 4\alpha^4 \sum_{i=1}^{d} \left|\phi(\beta_i) - \frac{\beta_i^2}{4\alpha^4}\right| \tag{70}$$

$$\leq \sum_{i=1}^{d} \beta_i^2 \cdot \frac{\beta_i^4 + 2\alpha^4 \beta_i^2}{96\alpha^8} \tag{71}$$

$$\leq \|\beta\|_2^2 \max_i \frac{\beta_i^4 + 2\alpha^4 \beta_i^2}{96\alpha^8} \tag{72}$$

Therefore, $\alpha \geq \sqrt{\|\beta\|_2}\left(1 + \epsilon^{-\frac{1}{4}}\right)$ ensures

$$(1 - \epsilon)\|\beta\|_2^2 \leq 4\alpha^4 Q(\beta/\alpha^2) \leq (1 + \epsilon)\|\beta\|_2^2 \tag{73}$$

$\square$

**Theorem 2.** *For any* $0 < \epsilon < d$

$$\alpha \leq \min\left\{(2(1+\epsilon)\|\boldsymbol{\beta}_{L1}^*\|_1)^{-\frac{2+\epsilon}{2\epsilon}}, \exp\left(-d/(\epsilon\|\boldsymbol{\beta}_{L1}^*\|_1)\right)\right\} \implies \|\hat{\boldsymbol{\beta}}_\alpha\|_1 \leq (1+\epsilon)\|\boldsymbol{\beta}_{L1}^*\|_1$$

$$\alpha \geq \sqrt{2(1+\epsilon)(1+2/\epsilon)\|\boldsymbol{\beta}_{L2}^*\|_2} \implies \|\hat{\boldsymbol{\beta}}_\alpha\|_2^2 \leq (1+\epsilon)\|\boldsymbol{\beta}_{L2}^*\|_2^2$$

*Proof.* We prove the $\ell_1$ and $\ell_2$ statements separately.

$\ell_1$ **approximation**  First, we will prove that $\left\|\hat{\beta}_\alpha\right\|_1 < (1 + 2\epsilon)\|\boldsymbol{\beta}_{L1}^*\|_1$. By Lemma 1, since $\alpha \leq \alpha_1\left(\frac{\epsilon}{2+\epsilon}, (1 + 2\epsilon)\|\boldsymbol{\beta}_{L1}^*\|_1, d\right)$, for all $\beta$ with $\|\beta\|_1 \leq (1 + 2\epsilon)\|\boldsymbol{\beta}_{L1}^*\|_1$ we have

$$\|\beta\|_1\left(1 - \frac{\epsilon}{2+\epsilon}\right) \leq \frac{\alpha^2}{\ln(1/\alpha^2)}Q(\beta/\alpha^2) \leq \|\beta\|_1\left(1 + \frac{\epsilon}{2+\epsilon}\right) \tag{74}$$

Let $\beta$ be such that $X\beta = y$ and $\|\beta\|_1 = (1 + 2\epsilon)\|\boldsymbol{\beta}_{L1}^*\|_1$. Then

$$\frac{\alpha^2}{\ln(1/\alpha^2)}Q(\beta/\alpha^2) \geq \left(1 - \frac{\epsilon}{2+\epsilon}\right)\|\beta\|_1 \tag{75}$$

$$= \left(1 - \frac{\epsilon}{2+\epsilon}\right)(1 + 2\epsilon)\|\boldsymbol{\beta}_{L1}^*\|_1 \tag{76}$$

$$\geq \frac{\left(1 - \frac{\epsilon}{2+\epsilon}\right)}{\left(1 + \frac{\epsilon}{2+\epsilon}\right)}(1 + 2\epsilon)\frac{\alpha^2}{\ln(1/\alpha^2)}Q(\boldsymbol{\beta}_{L1}^*/\alpha^2) \tag{77}$$

$$= \frac{1 + 2\epsilon}{1 + \epsilon}\frac{\alpha^2}{\ln(1/\alpha^2)}Q(\boldsymbol{\beta}_{L1}^*/\alpha^2) \tag{78}$$

$$> \frac{\alpha^2}{\ln(1/\alpha^2)}Q(\boldsymbol{\beta}_{L1}^*/\alpha^2) \tag{79}$$

$$\geq \frac{\alpha^2}{\ln(1/\alpha^2)}Q(\hat{\beta}_\alpha/\alpha^2) \tag{80}$$

Therefore, $\beta \neq \hat{\beta}_\alpha$. Furthermore, let $\beta$ be any solution $X\beta = y$ with $\|\beta\|_1 > (1 + 2\epsilon)\|\boldsymbol{\beta}_{L1}^*\|_1$. It is easily confirmed that there exists $c \in (0,1)$ such that the point $\beta' = (1 - c)\beta + c\boldsymbol{\beta}_{L1}^*$ is satisfies both $X\beta' = y$ and $\|\beta'\|_1 = (1 + 2\epsilon)\|\boldsymbol{\beta}_{L1}^*\|_1$. By the convexity of $Q$, this implies $Q(\beta/\alpha^2) \geq Q(\beta'/\alpha^2) > Q_\alpha(\hat{\beta}_\alpha/\alpha^2)$. Thus a $\beta$ with a large $\ell_1$ norm cannot be a solution, even if $\frac{\alpha^2}{\ln(1/\alpha^2)}Q(\beta/\alpha^2) \ngtr \|\beta\|_1$.

Since $\left\|\hat{\beta}_\alpha\right\|_1 < (1 + 2\epsilon)\|\boldsymbol{\beta}_{L1}^*\|_1$, we conclude

$$\left\|\hat{\beta}_\alpha\right\|_1 \leq \frac{1}{1 - \frac{\epsilon}{2+\epsilon}}\frac{\alpha^2}{\ln(1/\alpha^2)}Q(\hat{\beta}_\alpha/\alpha^2) \tag{81}$$

$$\leq \frac{1}{1 - \frac{\epsilon}{2+\epsilon}}\frac{\alpha^2}{\ln(1/\alpha^2)}Q(\boldsymbol{\beta}_{L1}^*/\alpha^2) \tag{82}$$

$$\leq \frac{1 + \frac{\epsilon}{2+\epsilon}}{1 - \frac{\epsilon}{2+\epsilon}}\|\boldsymbol{\beta}_{L1}^*\|_1 \tag{83}$$

$$= (1 + \epsilon)\|\boldsymbol{\beta}_{L1}^*\|_1 \tag{84}$$

First, we will prove that $\left\|\hat{\beta}_\alpha\right\|_2 < (1 + 2\epsilon)\|\boldsymbol{\beta}_{L2}^*\|_2$. By Lemma 3, since $\alpha \geq \alpha_2\left(\frac{\epsilon}{2+\epsilon}, (1 + 2\epsilon)\|\boldsymbol{\beta}_{L2}^*\|_2\right)$, for all $\beta$ with $\|\beta\|_2 \leq (1 + 2\epsilon)\|\boldsymbol{\beta}_{L2}^*\|_2$ we have

$$\|\beta\|_2^2\left(1 - \frac{\epsilon}{2+\epsilon}\right) \leq 4\alpha^4 Q(\beta/\alpha^2) \leq \|\beta\|_2^2\left(1 + \frac{\epsilon}{2+\epsilon}\right) \tag{85}$$

Let $\beta$ be such that $X\beta = y$ and $\|\beta\|_2 = (1 + 2\epsilon)\|\boldsymbol{\beta}_{L2}^*\|_2$. Then

$$4\alpha^4 Q(\beta/\alpha^2) \geq \left(1 - \frac{\epsilon}{2+\epsilon}\right)\|\beta\|_2^2 \tag{86}$$

$$= \left(1 - \frac{\epsilon}{2+\epsilon}\right)(1 + 2\epsilon)\|\boldsymbol{\beta}_{L2}^*\|_2^2 \tag{87}$$

$$\geq \frac{\left(1 - \frac{\epsilon}{2+\epsilon}\right)}{\left(1 + \frac{\epsilon}{2+\epsilon}\right)}(1 + 2\epsilon)4\alpha^4 Q(\boldsymbol{\beta}_{L2}^*/\alpha^2) \tag{88}$$

$$= \frac{1 + 2\epsilon}{1 + \epsilon}4\alpha^4 Q(\boldsymbol{\beta}_{L2}^*/\alpha^2) \tag{89}$$

$$> 4\alpha^4 Q(\boldsymbol{\beta}_{L2}^*/\alpha^2) \tag{90}$$

$$\geq 4\alpha^4 Q(\hat{\beta}_\alpha/\alpha^2) \tag{91}$$

Therefore, $\beta \neq \hat{\beta}_\alpha$. Furthermore, let $\beta$ be any solution $X\beta = y$ with $\|\beta\|_2 > (1 + 2\epsilon) \|\boldsymbol{\beta}_{L2}^*\|_2$. It is easily confirmed that there exists $c \in (0, 1)$ such that the point $\beta' = (1-c)\beta + c\boldsymbol{\beta}_{L2}^*$ satisfies $X\beta' = y$ and $\|\beta'\|_2 = (1 + 2\epsilon) \|\boldsymbol{\beta}_{L2}^*\|_2$. By the convexity of $Q$, this implies $Q(\beta/\alpha^2) \geq Q(\beta'/\alpha^2) > Q(\boldsymbol{\beta}_{L2}^*/\alpha^2)$. Thus a $\beta$ with a large $\ell_2$ norm cannot be a solution, even if $4\alpha^4 Q(\beta/\alpha^2) \not\approx \|\beta\|_2^2$.

Since $\left\|\hat{\beta}_\alpha\right\|_2 < (1 + 2\epsilon) \|\beta_2^*\|_2$, we conclude

$$\left\|\hat{\beta}_\alpha\right\|_2^2 \leq \frac{1}{1 - \frac{\epsilon}{2+\epsilon}} 4\alpha^4 Q(\hat{\beta}_\alpha/\alpha^2) \tag{92}$$

$$\leq \frac{1}{1 - \frac{\epsilon}{2+\epsilon}} 4\alpha^4 Q(\boldsymbol{\beta}_{L2}^*/\alpha^2) \tag{93}$$

$$\leq \frac{1 + \frac{\epsilon}{2+\epsilon}}{1 - \frac{\epsilon}{2+\epsilon}} \|\boldsymbol{\beta}_{L2}^*\|_2^2 \tag{94}$$

$$= (1 + \epsilon) \|\boldsymbol{\beta}_{L2}^*\|_2^2 \tag{95}$$

$\square$

## C  Proof of Theorem 3

**Lemma 4.** *For the $D$-homogeneous model* (8)*,*

$$\forall_t \left\| X^\top \int_0^t r(\tau)d\tau \right\|_\infty \leq \frac{\alpha^{2-D}}{D(D-2)}$$

*Proof.* For the order-$D$ unbiased model $\boldsymbol{\beta}(t) = \mathbf{w}_+^D - \mathbf{w}_-^D$, the gradient flow dynamics are

$$\dot{\mathbf{w}}_+(t) = -\frac{dL}{d\mathbf{w}_+} = -DX^\top r(t) \circ \mathbf{w}_+^{D-1}(t), \quad \mathbf{w}_+(0) = \alpha 1 \tag{96}$$

$$\implies \mathbf{w}_+(t) = \left(\alpha^{2-D}1 + D(D-2)X^\top \int_0^t r(\tau)d\tau\right)^{-\frac{1}{D-2}} \tag{97}$$

Where $\circ$ denotes elementwise multiplication, $r(t) = X\boldsymbol{\beta}(t) - y$, and where all exponentiation is elementwise. Similarly,

$$\dot{\mathbf{w}}_-(t) = -\frac{dL}{d\mathbf{w}_-} = DX^\top r(t) \circ \mathbf{w}_-^{D-1}(t), \quad \mathbf{w}_-(0) = \alpha 1 \tag{98}$$

$$\implies \mathbf{w}_-(t) = \left(\alpha^{2-D}1 - D(D-2)X^\top \int_0^t r(\tau)d\tau\right)^{-\frac{1}{D-2}} \tag{99}$$

First, we observe that $\forall_t \forall_i \, \mathbf{w}_+(t)_i \geq 0$ and $\forall_t \forall_i \, \mathbf{w}_-(t)_i \geq 0$. This is because at time 0, $\mathbf{w}_+(0)_i = \mathbf{w}_-(0)_i = \alpha > 0$; the gradient flow dynamics are continuous; and $\mathbf{w}_+(t)_i = 0 \implies \dot{\mathbf{w}}_+(t)_i = 0$ and $\mathbf{w}_-(t)_i = 0 \implies \dot{\mathbf{w}}_-(t)_i = 0$.

Consequently,

$$0 \leq \mathbf{w}_+(t)_i^{2-D} = \alpha^{2-D} + D(D-2)\left[X^\top \int_0^t r(\tau)d\tau\right]_i \tag{100}$$

$$0 \leq \mathbf{w}_-(t)_i^{2-D} = \alpha^{2-D} - D(D-2)\left[X^\top \int_0^t r(\tau)d\tau\right]_i \tag{101}$$

$$\implies -\alpha^{2-D} \leq D(D-2)\left[X^\top \int_0^t r(\tau)d\tau\right]_i \leq \alpha^{2-D} \tag{102}$$

which concludes the proof. $\square$

**Theorem 3.** *For any $\alpha$ and $D \geq 3$, if gradient flow reaches a solution $X\boldsymbol{\beta}_{\alpha,D}(\infty) = y$, then*

$$\boldsymbol{\beta}_{\alpha,D}(\infty) = \arg\min_{\boldsymbol{\beta}} Q_\alpha^D(\boldsymbol{\beta}) \ s.t. \ \mathbf{X}\boldsymbol{\beta} = \mathbf{y}$$

*where $Q_\alpha^D(\boldsymbol{\beta}) = \sum_{i=1}^d q_D(\boldsymbol{\beta}_i/\alpha^D)$ and $q_D = \int h_D^{-1}$ is the antiderivative of the unique inverse of $h_D(z) = (1-z)^{-\frac{D}{D-2}} - (1+z)^{-\frac{D}{D-2}}$ on $[-1,1]$. Furthermore, $\lim_{\alpha \to 0} \boldsymbol{\beta}_{\alpha,D}(\infty) = \boldsymbol{\beta}_{L1}^*$ and $\lim_{\alpha \to \infty} \boldsymbol{\beta}_{\alpha,D}(\infty) = \boldsymbol{\beta}_{L2}^*$.*

*Proof.* For the order-$D$ unbiased model $\boldsymbol{\beta}(t) = \mathbf{w}_+^D - \mathbf{w}_-^D$, the gradient flow dynamics are

$$\dot{\mathbf{w}}(t) = \frac{dL}{d\mathbf{w}} = -D\tilde{X}^\top r(t) \circ \mathbf{w}^{D-1}, \ \mathbf{w}(0) = \alpha 1$$

$$\implies \mathbf{w}(t) = \left(\alpha^{2-D} + D(D-2)\tilde{X}^\top \int_0^t r(\tau)d\tau\right)^{-\frac{1}{D-2}}$$

$$\implies \boldsymbol{\beta}(t) = \alpha^D \left(1 + \alpha^{D-2}D(D-2)X^\top \int_0^t r(\tau)d\tau\right)^{-\frac{D}{D-2}}$$

$$-\alpha^D \left(1 - \alpha^{D-2}D(D-2)X^\top \int_0^t r(\tau)d\tau\right)^{-\frac{D}{D-2}}$$

where $\tilde{X} = [X \ \ -X]$ and $r(t) = X\boldsymbol{\beta}(t) - y$. Supposing $\boldsymbol{\beta}(t)$ converges to a zero-error solution,

$$X\boldsymbol{\beta}(\infty) = \mathbf{y}\boldsymbol{\beta}(\infty) = \alpha^D h_D(X^\top \nu(\infty))$$

where $\nu(\infty) = -\alpha^{D-2}D(D-2)\int_0^\infty r(\tau)d\tau$ and the function $h_D$ is applied elementwise and is defined

$$h_D(z) = (1-z)^{-\frac{D}{D-2}} - (1+z)^{-\frac{D}{D-2}}$$

By Lemma 4, $\|X^\top \nu\|_\infty \leq 1$, so the domain of $h_D$ is the interval $[-1,1]$, upon which it is monotonically increasing from $h_D(-1) = -\infty$ to $h_D(1) = \infty$. Therefore, there exists an inverse mapping $h_D^{-1}(t)$ with domain $[-\infty, \infty]$ and range $[-1, 1]$.

This inverse mapping unfortunately does not have a simple closed form. Nevertheless, it is the root of a rational equation. Using the approach outlined in Appendix A (12), we conclude:

$$Q_\alpha^D(\boldsymbol{\beta}) = \sum_i \int_0^{\boldsymbol{\beta}_i/\alpha^D} h_D^{-1}(t)dt$$

**Rich regime** Next, we show that if gradient flow reaches a solution $X\boldsymbol{\beta}_{\alpha,D}(\infty) = y$, then $\lim_{\alpha \to 0} \boldsymbol{\beta}_{\alpha,D}(\infty) = \boldsymbol{\beta}_{L1}^*$ for any $D$. This is implied by the work of Arora et al. (2019a), but we include it here for an alternative, simpler proof for our special case, and for completeness's sake.

The KKT conditions for $\boldsymbol{\beta} = \boldsymbol{\beta}_{L1}^*$ are $X\boldsymbol{\beta} = y$ and $\exists \nu \ \text{sign}(\boldsymbol{\beta}) = X^\top \nu$ (where $\text{sign}(0) = [-1, 1]$). The first condition is satisfied by assumption. Define $\nu$ as above. We will demonstrate that the second condition holds too in the limit as $\alpha \to 0$.

First, by Lemma 4, $\|X^\top \nu\|_\infty \leq 1$ for all $\alpha$ and $D$. Thus, for any coordinates $i$ such that $\lim_{\alpha \to 0}[\boldsymbol{\beta}_{\alpha,D}(\infty)]_i = 0$, the second KKT condition holds. Consider now $i$ for which $\lim_{\alpha \to 0}[\boldsymbol{\beta}_{\alpha,D}(\infty)]_i > 0$. As shown above,

$$\lim_{\alpha \to 0}[\boldsymbol{\beta}_{\alpha,D}(\infty)]_i = \lim_{\alpha \to 0} \alpha^D \left(1 - [X^\top \nu]_i\right)^{-\frac{D}{D-2}} - \alpha^D \left(1 + [X^\top \nu]_i\right)^{-\frac{D}{D-2}} > 0 \quad (103)$$

$$\implies \lim_{\alpha \to 0} \alpha^D \left(1 - [X^\top \nu]_i\right)^{-\frac{D}{D-2}} > 0 \quad (104)$$

This and $[X^\top \nu]_i \leq 1$ implies $\lim_{\alpha \to 0}[X^\top \nu]_i = 1$, and thus the positive coordinates satisfy the second KKT condition. An identical argument can be made for the negative coordinates.

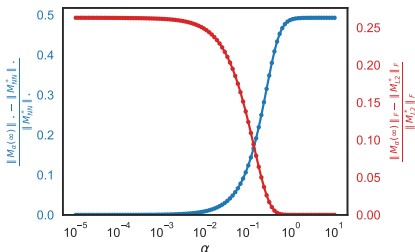

Figure 5: The plots demonstrate the regimes where gradient flow implicitly minimizes nuclear norm and $\ell_2$ norm, respectively.

**Kernel Regime**    Finally, we show that if gradient flow reaches a solution $X\boldsymbol{\beta}_{\alpha,D}(\infty) = y$, then $\lim_{\alpha\to\infty} \boldsymbol{\beta}_{\alpha,D}(\infty) = \boldsymbol{\beta}_{L2}^*$ for any $D$.

First, since $X$ and $y$ are finite, there exists a solution $\beta^*$ whose entries are all finite, and thus all the entries of $\boldsymbol{\beta}_{\alpha,D}(\infty)$, which is the $Q_\alpha^D$-minimizing solution, will be finite.

The KKT conditions for $\boldsymbol{\beta} = \boldsymbol{\beta}_{L2}^*$ are $X\boldsymbol{\beta} = y$ and $\exists\mu \; \boldsymbol{\beta} = X^\top\mu$. The first condition is satisfied by assumption. Defining $\nu$ as above, we have

$$\lim_{\alpha\to\infty} [\boldsymbol{\beta}_{\alpha,D}(\infty)]_i = \lim_{\alpha\to 0} \alpha^D \left(1 - [X^\top\nu]_i\right)^{-\frac{D}{D-2}} - \alpha^D \left(1 + [X^\top\nu]_i\right)^{-\frac{D}{D-2}} < \infty \tag{105}$$

$$\implies \lim_{\alpha\to\infty} [X^\top\nu]_i = 0 \tag{106}$$

Consequently, defining $\mu = \frac{2D\alpha^D}{D-2}\nu$, and observing that for small $z$,

$$(1-z)^{-\frac{D}{D-2}} - (1+z)^{-\frac{D}{D-2}} = \frac{2D}{D-2}z + O(z^3) \tag{107}$$

we conclude

$$\lim_{\alpha\to\infty} \frac{[\boldsymbol{\beta}_{\alpha,D}(\infty)]_i}{[X^\top\mu]_i} = \lim_{\alpha\to 0} \frac{\alpha^D \left(1 - [X^\top\nu]_i\right)^{-\frac{D}{D-2}} - \alpha^D \left(1 + [X^\top\nu]_i\right)^{-\frac{D}{D-2}}}{[X^\top\mu]_i} \tag{108}$$

$$= \lim_{\alpha\to 0} \frac{\alpha^D \left(\frac{2D}{D-2}[X^\top\nu]_i + O([X^\top\nu]_i^3)\right)}{\frac{2D\alpha^D}{D-2}[X^\top\nu]_i} \tag{109}$$

$$= 1 + \lim_{\alpha\to 0} O([X^\top\nu]_i^2) \tag{110}$$

$$= 1 \tag{111}$$

Thus, the KKT conditions are satisfied for $\lim_{\alpha\to\infty} \boldsymbol{\beta}_{\alpha,D}(\infty) = \boldsymbol{\beta}_{L2}^*$.  □

## D  MATRIX EXPERIMENTS

In the appendix, we provide additional results similar to those in Section 6. First, in Figure 5, we plot the implicit regularization behavior of gradient flow limits with identity initialization as in Figure 3(a): in "rich" regime (small $\alpha$) we recover the minimum nuclear norm solution $M_{NN}^* = \arg\min_{P_\Omega(M)=y} \|M\|_\star$, while "kernel" regime recovers the minimum Frobenius norm solution $M_{L@}^* = \arg\min_{P_\Omega(M)=y} \|M\|_2$

## E  NEURAL NETWORK EXPERIMENTS

**Synthetic Experiments**    We construct a synthetic training set with $N = 10$ points drawn uniformly from the unit circle in $\mathbb{R}^2$ and labelled by a teacher model with 1 hidden layer of 3 units. We train fully connected ReLU networks with depths 2, 3, and 5 with 30 units per layer to minimize the square loss using full gradient descent with constant stepsize 0.01 until the training loss is below $10^{-9}$. We use Uniform He initialization for the weights and then multiply them by $\alpha$.

Here, we describe the details of the neural network implementations for the MNIST and CIFAR10 experiments.

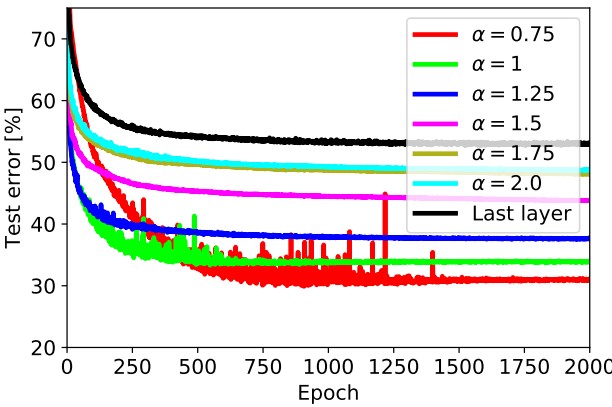

Figure 6: Training curves for the CIFAR10 experiments

**MNIST** Since our theoretical results hold for the squared loss and gradient flow dynamics, here we empirically assess whether different regimes can be observed when training neural networks following standard practices.

We train a fully-connected neural network with a single hidden layer composed of 5000 units on the MNIST dataset, where weights are initialized as $\alpha \mathbf{w}_0$, $\mathbf{w}_0 \sim \mathcal{N}\left(0, \sqrt{\frac{2}{n_{in}}}\right)$, $n_{in}$ denoting the number of units in the previous layer, as suggested by He et al. (2015). SGD with a batch size of 256 is used to minimize the cross-entropy loss over the 60000 training points, and error over the 10000 test samples are used as measure of generalization. For each value of $\alpha$, we search over learning rates $(0.5, 0.01, 0.05, \dots)$ and use the one which resulted in best generalization.

There is a visible phase transition in Figure 4e in terms of generalization ($\approx 1.4\%$ error for $\alpha \leq 2$, and $\approx 2.4\%$ error for $\alpha \geq 50$), even though every network reached 100% training accuracy and less than $10^{-5}$ cross-entropy loss. The black line indicates the test error (2.7%) when training only the output layer of the network, as a proxy for the performance of a linear predictor with features given by a fixed, randomly-initialized hidden layer.

**CIFAR10** We trained a VGG11-like architecture, which is as follows: 64-M-128-M-256-256-M-512-512-M-512-512-M-FC (numbers represent the number of channels in a convolution layers with no bias, M is a maxpooling layer, and FC is a fully connected layer). Weights were initialized using Uniform He initialization multiplied by $\alpha$. No data augmentation was used, and training done using SGD with batch size of 128 and learning rate of 0.0001. All experiments ran for 2000 epochs, and reached 100% train accuracy except when training only the last layer, which reached 50.38% train accuracy with LR = 0.001 (chosen after hyperparameter tuning).

In addition, to approximate the test error in the kernel regime, we experimented with freezing the bottom layers and only training the output layer for both datasets (the solid lines in Figures 4e and 4f).

Figure 6 illustrates some of the optimization difficulties that arise from using smaller $\alpha$ as discussed in Section 4.

