# OpenReview forum: "Kernel and Rich Regimes in Overparametrized Models"
_ICLR.cc/2020/Conference — Reject_

### Official Review · AnonReviewer2 · 2019-10-22
**Official Blind Review #2**

**Rating:** 8

**Review:**

I really appreciated this paper. It discusses a very complex question ("Are we learning in a kernel regime, or in a rich regime where features are identified") by looking at perhaps the simplest model the authors could think of, and then study in detail the model. And how simple it turns out to be: just a linear regression with a twist. All in all, the paper is indeed is a clear demonstration that the differences between"Kernel" regime and one where some actual Learning is done can be demonstrated on simple examples. It is also the simplest model where one can observe  a non-trivial inductive bias and 'implicit regularisation'

I do not have much to say on the paper, except that I fully support publication.





**Experience Assessment:**

I have read many papers in this area.

**Review Assessment: Checking Correctness Of Derivations And Theory:**

I assessed the sensibility of the derivations and theory.

**Review Assessment: Checking Correctness Of Experiments:**

I did not assess the experiments.

**Review Assessment: Thoroughness In Paper Reading:**

I read the paper at least twice and used my best judgement in assessing the paper.

---

> ### Author Response · Authors · 2019-11-12
> **Thank you for your comments!**
>
> .

---

### Official Review · AnonReviewer1 · 2019-10-23
**Official Blind Review #1**

**Rating:** 6

**Review:**

This paper analyzes an inductive bias of the gradient flow for diagonal two-or higher-homogeneous models and characterizes a limit point depending on the initialization scale of parameters. Concretely, the paper shows that the gradient flow converges to an interpolator attaining minimum L1- (or L2-norm) when the scale is small (or large). In addition, these analyses are well verified empirically on MNIST and CIFAR-10 datasets.

Quality:
The work is of good quality and is technically sound.

Clarity:
The paper is well organized and easy to read.

Significance:
To explain the generalization ability of powerful machine learning models that can perfectly learn a training dataset, the implicit bias of the optimization methods and models play key roles when explicit regularization is not adopted. For instance, deep neural networks fall into this scenario. I think this paper makes a better contribution in this line of researches. Although, homogeneous models treated in this study is restricted (essentially linear models) and a theory is limited to the continuous gradient flow, these settings are rather common in this context. In [Gunasekar+(2017)], the convergence to the minimum L1-norm solution was shown for a slightly different model when the scale goes to zero. However, in addition to this property, the paper analyzes arbitrary scales of parameters and shows the convergence to the minimum L2-norm solution when the scale goes to infinity for diagonal homogeneous models.
It would be nice if the authors could emphasize the technical difficulty compared to [Gunasekar+(2017)] to strengthen the contribution of the paper.

A few questions:
- Can this analysis be extended to the setting of early stopping? Toward a better explanation of the generalization performance of deep learning, understanding of the inductive bias of the early stopping before convergence is more important.
- A provided theory is limited to linear models essentially. Is it possible to extend a theory to non-linear models?

-----
Update:
I thank the authors for the response. I am convinced of the difference from [Gunasekar+(2017)] and my review stands. I  would like to keep my score.

**Experience Assessment:**

I have read many papers in this area.

**Review Assessment: Checking Correctness Of Derivations And Theory:**

I assessed the sensibility of the derivations and theory.

**Review Assessment: Checking Correctness Of Experiments:**

I assessed the sensibility of the experiments.

**Review Assessment: Thoroughness In Paper Reading:**

I made a quick assessment of this paper.

---

> ### Author Response · Authors · 2019-11-12
> **Response to reviewer #1**
>
> Thank you for your comments.
>
> Regarding the technical difficulty and proof technique compared to [Gunasekar+2017]:
>
> The main novelty here is that we start with an optimization procedure (finite init + grad flow) and work backwards from its dynamics to uncover de-novo a complexity measure it is implicitly minimizing.  This is different from [Gunasekar+2017] that started with a guess about what might be minimized (namely the nuclear norm) and worked forward from it.  We should have indeed made this more explicit, and following your comment, we updated the manuscript explaining our derivation—-see the discussion before the proof of Theorem 1 on page 11 in the updated manuscript.  As detailed there, we start from the dynamics, work out constraints satisfied by the dynamics, relate them to KKT conditions for a minimization problem of an unknown implied regularizer Q, and from that obtain a differential equation on Q which we solve.  In a sense, the approach of [Gunasekar+2017] was “guess-and-check,” whereas in this paper we developed a principled approach that allows us to calculate the implied regularizer when we don’t have an obvious guess.
>
> Understanding the behavior in other settings (e.g. gradient descent instead of gradient flow,  early stopping instead of running until convergence, etc.) is certainly interesting and important to understanding what happens in practical settings. In our setting, we do not know exactly what happens with early stopping. One reasonable hypothesis would be that optimizing with gradient flow from initialization alpha with early stopping would reach a point on the Q_alpha regularization path. However, we know that this is NOT necessarily the case.
>
> We also have preliminary results that show that for the same simple model we consider in Section 4 with separable data and with the logistic or exponential loss (versus the square loss), then early stopping and the implicit bias are inseparable from each other. In particular, for any given initialization, the predictor will eventually converge in some sense to the minimum L1 margin predictor. On the other hand, for any given early stopping time, when the initialization is large enough, gradient flow will reach the maximum L2 margin predictor at the early stopping time. Therefore, the implicit bias depends simultaneously on the early stopping time AND the scale of the initialization. There are lots of interesting questions to explore here, which we hope to answer in the future.
>
> In the case of non-linear models, it is much harder to characterize the implicit bias of the model outside of the kernel regime (in which case the non-linear model is effectively linear). Our experimental results in Figure 4 suggest that a similar phenomenon is occurring, where a small scale of initialization allows for better test error than larger initialization. We suspect that this corresponds to some sort of “rich regime” for the non-linear models we use in the experiments, although we can’t say exactly what implicit bias that corresponds to. Extending the understanding from linear to non-linear models is a very important question for future work.

---

### Official Review · AnonReviewer3 · 2019-10-26
**Official Blind Review #3**

**Rating:** 3

**Review:**

This paper investigates the two regimes in the training of overparameterized networks (with small learning rates):
* kernel regime: the tangent kernel doesn't change much during training. The training behavior is then well approximated by a linear model (Taylor expansion at the initialization). This can happen when the weights are initialized to large values.
* rich regime: The kernel regime is turned into a rich regime when the assumptions of kernel regimes aren't met.

Specifically, the paper emphasizes how the scale of initialization controls the transition between the two regimes, which was first pointed out by Chizat & Bach (2018).

My main concern is that it is unclear what unique contributions are made by the paper, as the theoretical results are not more general than that of Chizat & Bach (2018). The contributions are not clearly stated and I can only see the execution of ideas from Chizat & Bach (2018) and applying them to more concrete examples, which leads to analytical results (for linear networks) in Theorem 1/2. This feels rather incremental.

Some other comments:
* In experiments it was shown that popular initialization schemes are right on the edge of entering the kernel regime, which is very interesting. How does this change with network widths and different architectures?
* It's difficult to see what Figure 2b tells because several notations are undefined. What are $e_1$ and $1_d$?


**Experience Assessment:**

I have read many papers in this area.

**Review Assessment: Checking Correctness Of Derivations And Theory:**

I assessed the sensibility of the derivations and theory.

**Review Assessment: Checking Correctness Of Experiments:**

I assessed the sensibility of the experiments.

**Review Assessment: Thoroughness In Paper Reading:**

I read the paper at least twice and used my best judgement in assessing the paper.

---

> ### Author Response · Authors · 2019-11-12
> **Response to reviewer #3**
>
> Thank you for your comments.
>
> The main contribution over Chizat and Bach is working out the entire entire transition between the kernel and rich regimes as a function of the initialization.  Chizat and Bach only describe the limiting behaviour when alpha->infty, while we get a precise description for every finite alpha.  Beyond showing an example of exactly how the Chizat and Bach limit comes about (which we think is also valuable in and of itself), understanding the behavior as a function of alpha (and not only in the limit alpha->infty) provided for new insights that were not apparent in previous work:
>
> - We show that reaching the rich regime can be very slow, and exponentially small initialization is necessary to enter this limit. This provides an explanation for why it has often been difficult to demonstrate the rich regime empirically, and can help cast discussions on this limit in new light. For example, [Arora+2019] cast doubt on the hypothesis that the rich regime in matrix factorization corresponds to nuclear norm minimization on the basis of several experiments where the nuclear norm is not exactly minimized. However, our work suggests that this may be explained by the fact that the initialization used isn’t (and in a sense can’t be) small enough, and gives a different twist to their results: maybe in the limit we do get nuclear norm, but the behavior before the limit is important since its extremely difficult to reach this limit.
>
> - Our analysis highlights the importance of the transition regime, between the two extremes (see discussion in the final two paragraphs of page 5). E.g., for the sparse regression problem described in Section 4, although the rich regime will lead to better learning, it is more difficult from an optimization perspective, and the “correct” initialization to use is in the transition. Our neural network experiments (see Figure 4) provide further evidence, showing that standard successful initialization schemes correspond to a point right on the boundary of leaving behind the good generalization of the rich regime and entering the kernel regime.
>
> - We connect between the scale of initialization and the sample complexity of learning (see Figures 1c and 2c). This level of detail was not explored by previous asymptotic analyses, and it further reinforces the importance of being in the transition regime as described above.
>
> - We investigate how higher order models/depth relate to the transition. In Section 5, we show that order-3+ models have rich regime behavior with dramatically larger initialization compared to the order-2 model. Similarly, in Figure 4a,b we see that deeper models have good rich regime generalization behavior at larger initializations than shallower models.
>
> - Finally, we develop an approach for deriving the implicit bias for a particular method in situations where it is unclear a priori what the implicit bias will be. See our response to Reviewer #1 and the discussion preceding the proof of Theorem 1 on pg 11 in the updated manuscript.
>
> All the above insights rely on understanding the behavior as a function of alpha, and are not possible when only considering the alpha->infty endpoint as in Chizat and Bach.  Beyond these specific insights, we expect our detailed description and the methodology developed (which is entirely different from Chizat and Bach) will also serve as a basis for future investigation.
>
> In summary, our work is obviously heavily influenced by Chizat and Bach, but we take their work as a starting point and go well beyond what they already analyzed.
>
> For all three of the experiments in Figure 4, which each use different architectures, we see that alpha ~ 1 has good test error and slightly larger alpha starts to degrade performance. This suggests that this phenomenon is fairly consistent over different architectures, although we did not experiment with different widths explicitly. This would be an interesting experiment to conduct, although we suspect that it is no coincidence that the transition point occurs right around alpha = 1, and thus we expect our experiments to be fairly robust to changes in width.
>
> Notation: e_1=[1,0,0,0,...,0] is the first standard basis vector and 1_d=[1,1,1,...,1] is the vector of all-ones in R^d.  We have added an explanation in the figure caption (page 6).

---

### Decision · Program_Chairs · 2019-12-19

**Decision:**

Reject

**Comment:**

The paper studies how the size of the initialization of neural network weights affects whether the resulting training puts the network in a "kernel regime" or a "rich regime". Using a two-layer model they show, theoretically and practically, the transition between kernel and rich regimes. Further experiments are provided for more complex settings.

The scores of the reviewers were widely spread, with a high score (8) from a low confidence reviewer with a very short review. While the authors responded to the reviewer comments, two of the reviewers (importantly including the one recommending reject) did not further engage.

Overall, the paper studies an important problem, and provides insight into how weight initialization size can affect the final network. Unfortunately, there are many strong submissions to ICLR this year, and the submission in its current state is not yet suitable for publication.